# Bridging Neural Learning and Symbolic Reasoning: A Differentiable Fuzzy Description Logic Framework

## Abstract

Injecting logical reasoning as a structural prior into deep learning models is a central goal of Neural-Symbolic AI. A key challenge is making expressive formalisms like the Description Logic (DL) $\mathcal{ALC}$ compatible with gradient-based optimization without sacrificing their rich semantics. While the theory of fuzzy $\mathcal{ALC}$ offers a path, a significant methodological gap has persisted: although operator-agnostic fuzzy reasoning has been explored in first-order logic-based frameworks, these methods face fundamental computational barriers when scaling to real-world ontologies, leaving the choice of fuzzy operators for DLs largely unexplored. This paper introduces NeSy$\mathcal{ALC}$, an end-to-end learning framework designed to bridge this gap by transforming fuzzy $\mathcal{ALC}$ from a theoretical construct into a practical tool for representation learning at scale. Our framework is operator-agnostic within the DL setting, allowing us to conduct the first systematic empirical analysis of how different t-norms and fuzzy implications impact learning on diverse knowledge bases. We find that no single operator is universally optimal, motivating our second contribution: a novel adaptive dual-loss optimization strategy that dynamically adjusts its objective based on the logical structure of the knowledge base, enhancing learning robustness. Through extensive experiments on ontology completion and semantic image interpretation tasks, we demonstrate that NeSy$\mathcal{ALC}$ consistently and statistically significantly outperforms established baselines. Our work operationalizes fuzzy $\mathcal{ALC}$ for modern machine learning, providing a practical and robust framework for injecting rich symbolic knowledge into neural models.

## 1 Introduction

Neural networks, despite their remarkable success in pattern recognition, often operate in a logical vacuum, leading to predictions that violate fundamental real-world constraints (Marcus, 2020). Consider a state-of-the-art model for histopathology analyzing a tissue sample: it might confidently classify the same region as both adenocarcinoma and squamous cell carcinoma—a biological impossibility, as these are mutually exclusive tumor types arising from distinct cellular lineages (Travis et al., 2015). Such failures are not isolated; in materials science, models may predict thermodynamically incompatible atomic structures (Bartel, 2022), and in autonomous systems, actions may violate strict safety protocols (Sun et al., 2019). These examples illustrate a broader gap between perceptual pattern-matching and structured reasoning.

The field of Neural-Symbolic (NeSy) AI seeks to bridge this gap by integrating neural learning with the formal rigor of symbolic logic (d'Avila Garcez et al., 2019). A central challenge in this integration is choosing an appropriate logical formalism. Full First-Order Logic (FOL) (Ben-Ari, 2012), while expressive—thereby providing a rich inductive bias for modeling complex domain knowledge—is undecidable (Church, 1936; Turing, 1936), and practical systems based on it must often relax semantics to achieve tractability. The Description Logic $\mathcal{ALC}$ (Schmidt-Schauß & Smolka, 1991) offers an appealing alternative: it retains high expressivity—supporting full Boolean connectives and quantifiers—while remaining EXPTIME-complete and thus decidable. Building on $\mathcal{ALC}$ allows us to design a learning framework with **provable polynomial-time complexity per training iteration**, ensuring both semantic fidelity and scalability on production-scale knowledge bases.

The idea of making $\mathcal{ALC}$ differentiable via fuzzy semantics is not new; indeed, Straccia (2001) laid the theoretical groundwork for reasoning in fuzzy $\mathcal{ALC}$ over two decades ago. In parallel, foundational frameworks such as Logic Tensor Networks (LTN) (Badreddine et al., 2022) and Differentiable Fuzzy Logics (DFL) (van Krieken et al., 2022) have advanced the study of differentiable fuzzy operators within FOL, establishing an operator-agnostic paradigm that decouples logical structure from the choice of fuzzy semantics. However, these FOL-based frameworks do not directly address the unique computational demands of Description Logics (DLs) (Baader et al., 2017): DL reasoning is inherently terminological, operating over concept hierarchies and role restrictions rather than arbitrary predicates, and must scale to industrial ontologies containing millions of axioms. Bridging this gap—translating the flexibility of operator-agnostic fuzzy reasoning into a DL-native, scalable architecture—remains an open challenge.

This paper introduces NeSy$\mathcal{ALC}$, a framework designed to bridge this gap. We transform fuzzy $\mathcal{ALC}$ from a theoretical construct into a practical and powerful NeSy tool by proposing the first learning framework that is both systematically flexible and adaptive. Our contributions are:

- We design and implement an end-to-end learning framework for fuzzy $\mathcal{ALC}$ that is **agnostic to the choice of underlying fuzzy operators**. By extending the operator-agnostic paradigm of prior FOL-based works to DLs, our framework enables a principled exploration of how different logical priors shape learned representations under DL-specific structural constraints.
- We conduct the first **systematic empirical analysis** of a wide range of fuzzy t-norms and implications for $\mathcal{ALC}$ reasoning. Our results reveal that no single operator is universally optimal, highlighting the critical impact of operator choice on learning outcomes.
- We introduce a novel **adaptive dual-loss optimization strategy** that dynamically adjusts its objective based on the logical structure of the input ontology. This addresses **representational collapse**—where models satisfy axioms vacuously without learning meaningful concepts—yielding a more robust learning process that reduces reliance on heuristic hyperparameter tuning.
- Through extensive experiments on ontology completion and semantic image interpretation, we demonstrate that our framework consistently and statistically significantly outperforms established NeSy baselines in recovering information from noisy and incomplete knowledge bases.

## 2 PRELIMINARIES: FROM CLASSICAL TO FUZZY DESCRIPTION LOGIC

Description logics (DLs) are decidable fragments of FOL designed to formalize terminological knowledge (Baader et al., 2017). Among the DL family, $\mathcal{ALC}$ (Attributive Concept Language with Complements) serves as a canonical representative, providing essential Boolean connectives and quantifiers while maintaining EXPTIME-complete decidability (Schmidt-Schauß & Smolka, 1991).

### 2.1 CLASSICAL $\mathcal{ALC}$ AND DECIDABILITY

Let $N_C$, $N_R$, and $N_I$ be pairwise disjoint and countably infinite sets of concept names, role names, and individual names, respectively. The syntax of $\mathcal{ALC}$-concepts is inductively defined by:

$$C, D \rightarrow \top \mid \bot \mid A \mid \neg C \mid C \sqcap D \mid C \sqcup D \mid \exists r.C \mid \forall r.C$$

where $A \in N_C$, $r \in N_R$, and $C, D$ range over concepts. A concept is *atomic* if it is a concept name and *complex* otherwise.

An $\mathcal{ALC}$ ontology $\mathcal{O}$ consists of a TBox $\mathcal{T}$ and an ABox $\mathcal{A}$. An $\mathcal{ALC}$-TBox is a finite set of axioms: concept inclusions $C \sqsubseteq D$ and concept equivalences $C \equiv D$ (abbreviating $C \sqsubseteq D$ and $D \sqsubseteq C$). Disjointness between concepts $C$ and $D$ is expressed as $C \sqcap D \sqsubseteq \bot$. An $\mathcal{ALC}$-ABox is a finite set of concept assertions $a : C$ and role assertions $(a, b) : r$, where $C$ is a concept, $r \in N_R$, and $a, b \in N_I$. The signature $\text{sig}(\mathcal{O})$ is the set of names occurring in $\mathcal{O}$.

The semantics is defined using an interpretation $\mathcal{I} = \langle \Delta^{\mathcal{I}}, \cdot^{\mathcal{I}} \rangle$, where $\Delta^{\mathcal{I}}$ is the domain (nonempty set), and $\cdot^{\mathcal{I}}$ is the interpretation function that maps: (i) each individual $a$ to an element $a^{\mathcal{I}} \in \Delta^{\mathcal{I}}$, (ii) each concept name $A$ to a subset $A^{\mathcal{I}} \subseteq \Delta^{\mathcal{I}}$, and (iii) each role name $r$ to a binary relation $r^{\mathcal{I}} \subseteq \Delta^{\mathcal{I}} \times \Delta^{\mathcal{I}}$. Essentially, the interpretation function $\mathcal{I}$ maps symbolic primitives into a vector space, forming the very representations we aim to learn. Complex concepts are interpreted as:

$$\top^{\mathcal{I}} = \Delta^{\mathcal{I}}, \ \bot^{\mathcal{I}} = \emptyset, \ (\neg C)^{\mathcal{I}} = \Delta^{\mathcal{I}} \setminus C^{\mathcal{I}}, \ (C \sqcap D)^{\mathcal{I}} = C^{\mathcal{I}} \cap D^{\mathcal{I}}, \ (C \sqcup D)^{\mathcal{I}} = C^{\mathcal{I}} \cup D^{\mathcal{I}},$$

$$(\exists r.C)^{\mathcal{I}} = \{a \in \Delta^{\mathcal{I}} \mid \exists b.(a, b) \in r^{\mathcal{I}} \land b \in C^{\mathcal{I}}\}, \ (\forall r.C)^{\mathcal{I}} = \{a \in \Delta^{\mathcal{I}} \mid \forall b.(a, b) \in r^{\mathcal{I}} \rightarrow b \in C^{\mathcal{I}}\}$$

An interpretation $\mathcal{I}$ satisfies $C \sqsubseteq D$ (written $\mathcal{I} \models C \sqsubseteq D$) iff $C^{\mathcal{I}} \subseteq D^{\mathcal{I}}$; $a : C$ iff $a^{\mathcal{I}} \in C^{\mathcal{I}}$; and $(a, b) : r$ iff $(a^{\mathcal{I}}, b^{\mathcal{I}}) \in r^{\mathcal{I}}$. $\mathcal{I}$ is a model of $\mathcal{O}$ (written $\mathcal{I} \models \mathcal{O}$) iff it satisfies every axiom in $\mathcal{O}$. An axiom $\beta$ is entailed by $\mathcal{O}$ (written $\mathcal{O} \models \beta$) iff $\beta$ is satisfied in every model of $\mathcal{O}$.

## 2.2 ZADEH'S FUZZY $\mathcal{ALC}$

Fuzzy set theory, proposed by Zadeh (Zadeh, 1965), manages imprecise knowledge through membership functions $\mu_X : U \to [0, 1]$, where $\mu_X(u)$ represents the truth value of '$u$ is $X$'. T-norms ($\otimes$), t-conorms ($\oplus$), and negation ($\ominus$) generalize classical operations (Klement et al., 2000).

**Definition 2.1 (Fuzzy Interpretation).** Following Straccia (Straccia, 2001), a fuzzy interpretation $\mathcal{I} = \langle \Delta^{\mathcal{I}}, \cdot^{\mathcal{I}} \rangle$ maps: (i) individuals $a$ to elements $a^{\mathcal{I}} \in \Delta^{\mathcal{I}}$, (ii) concept names $C$ to fuzzy sets $C^{\mathcal{I}} : \Delta^{\mathcal{I}} \to [0, 1]$, and (iii) role names $r$ to fuzzy relations $r^{\mathcal{I}} : \Delta^{\mathcal{I}} \times \Delta^{\mathcal{I}} \to [0, 1]$.

**Example 2.1.** Consider a domain $\Delta = \{p_1, p_2\}$ representing two patients. A fuzzy interpretation of the concept *Diabetic* might be *Diabetic*$^{\mathcal{I}} = [0.9, 0.2]$, meaning patient $p_1$ has a membership degree of 0.9 to the concept (highly diabetic), while $p_2$ has 0.2 (unlikely diabetic).

Complex concepts are interpreted inductively:

$$\top^{\mathcal{I}}(a) = 1, \quad \bot^{\mathcal{I}}(a) = 0, \quad (\neg C)^{\mathcal{I}}(a) = 1 - C^{\mathcal{I}}(a),$$

$$(C \sqcap D)^{\mathcal{I}}(a) = \min\{C^{\mathcal{I}}(a), D^{\mathcal{I}}(a)\}, \quad (C \sqcup D)^{\mathcal{I}}(a) = \max\{C^{\mathcal{I}}(a), D^{\mathcal{I}}(a)\},$$

$$(\exists r.C)^{\mathcal{I}}(a) = \sup_{b \in \Delta^{\mathcal{I}}} \left\{ \min \left\{ r^{\mathcal{I}}(a, b), C^{\mathcal{I}}(b) \right\} \right\}, \quad (\forall r.C)^{\mathcal{I}}(a) = \inf_{b \in \Delta^{\mathcal{I}}} \left\{ \max \left\{ 1 - r^{\mathcal{I}}(a, b), C^{\mathcal{I}}(b) \right\} \right\}$$

A Zadeh-$\mathcal{ALC}$ TBox contains fuzzy inclusions $C \sqsubseteq D$, satisfied in $\mathcal{I}$ iff $\forall a \in \Delta^{\mathcal{I}}, C^{\mathcal{I}}(a) \leq D^{\mathcal{I}}(a)$. Two concepts are fuzzy equivalent ($C \equiv D$) if $C^{\mathcal{I}}(a) = D^{\mathcal{I}}(a)$ for all $a$. An ABox contains fuzzy assertions $(a : C) \triangleright n$ or $((a, b) : r) \triangleright n$ where $\triangleright \in \{\geq, >, \leq, <\}$ and $n \in [0, 1]$. For simplicity, we write $\varphi = \{\phi \triangleright n\}$ and restrict $\triangleright$ to $=$ using pairs of inequalities. $\mathcal{I}$ is a model of fuzzy ontology $\mathcal{O}$ (written $\mathcal{I} \approx \mathcal{O}$) iff it satisfies each axiom.

Having established these foundations, we now describe how NeSy$\mathcal{ALC}$ operationalizes these semantics for gradient-based learning by overcoming the non-differentiability of classical operators ($\min, \max, \sup, \inf$).

# 3 NESY$\mathcal{ALC}$: A DIFFERENTIABLE FUZZY $\mathcal{ALC}$ FRAMEWORK

NeSy$\mathcal{ALC}$ operationalizes fuzzy $\mathcal{ALC}$ for gradient-based learning by adapting it with a suite of fully differentiable logical operators. The key challenge is reconciling continuous neural outputs with symbolic logical constraints. We address this through a structured framework: first, we define a differentiable interpretation of the logic; second, we design a modular loss objective by normalizing complex axioms; and third, we introduce three distinct optimization strategies to learn representations that satisfy the logical theory.

## 3.1 DIFFERENTIABLE INTERPRETATION FRAMEWORK

We start by defining how symbolic primitives are mapped into a continuous space suitable for learning. For a given ontology $\mathcal{O}' = \langle \mathcal{T}', \mathcal{A}' \rangle$, the ABox $\mathcal{A}'$ is treated as a set of fuzzy assertions where each fact $\phi \in \mathcal{A}'$ is initially assigned a truth value of 1.

The core of our framework is the interpretation $\mathcal{I} = \langle \Delta^{\mathcal{I}}, \cdot^{\mathcal{I}} \rangle$, defined over a domain $\Delta = N_I$ where each individual corresponds to a unique domain element. The interpretation function $\cdot^{\mathcal{I}}$ acts as a trainable embedding layer: (i) **Concept names** $C$ are mapped to vectors $C^{\mathcal{I}} \in [0, 1]^{|\Delta|}$. The $i$-th component, $C_i^{\mathcal{I}}$, represents the membership degree of the $i$-th individual to $C$; (2) **Role names** $r$ are mapped to matrices $r^{\mathcal{I}} \in [0, 1]^{|\Delta| \times |\Delta|}$. The entry $r_{ij}^{\mathcal{I}}$ represents the degree to which the ordered pair of individuals $(\Delta_i, \Delta_j)$ belongs to $r$.

These vectors and matrices are the trainable parameters of our model, forming the representations that are iteratively updated to satisfy the logical constraints of the TBox.

### 3.2 LEARNING WITH DIFFERENTIABLE LOGICAL LOSSES

With our representations defined, the central task is to design a loss function that guides the learning process. Unlike NeSy approaches that add logical constraints as regularization to supervised learning tasks (i.e., $\mathcal{L}_{task} + \lambda\mathcal{L}_{logic}$), NeSy$\mathcal{ALC}$ treats the logical knowledge base itself as the **primary and sole learning objective**. The goal is to learn fuzzy interpretations $(C^{\mathcal{I}}, r^{\mathcal{I}})$ that satisfy the axioms, rather than to improve a separate neural task with logical guidance. Consequently, our total loss $\mathcal{L}_{total}$ is the complete loss function, with no additional task-specific loss to balance. The design of this function hinges on two key components: the choice of mathematical operators to represent logical connectives, and a method to handle the syntactic complexity of arbitrary logical axioms.

#### 3.2.1 ON THE CHOICE OF LOGICAL OPERATORS

The translation of logical axioms into a differentiable loss is governed by the choice of fuzzy operators. This choice is non-trivial, as it involves a trade-off between operators with strong theoretical properties and those with favorable gradient properties for learning. Our framework is designed to be operator-agnostic, allowing for a systematic evaluation of:

**T-norms and T-conorms**    for conjunction ($\otimes$) and disjunction ($\oplus$). We implement and evaluate a suite of t-norms beyond the standard **Gödel t-norm**, including **Product**, **Łukasiewicz**, and the parameterized **Yager** and **Hamacher** families (van Krieken et al., 2022), to understand their impact on the optimization landscape.

**Fuzzy Implications**    for subsumption axioms ($C \sqsubseteq D$). We evaluate three primary types derived from the chosen t-norm/t-conorm: (i) **S-implications (Strong):** Defined as $I_S(x, y) = (1 - x) \oplus y$; (ii) **R-implications (Residuated):** Defined as $I_R(x, y) = \sup\{z \in [0, 1] \mid x \otimes z \leq y\}$, which often better model crisp entailment; (iii) **Sigmoidal Implications:** Defined by transforming standard implications with a sigmoid function to smooth gradients (van Krieken et al., 2022).

Our framework allows us to flexibly **pair a specific t-norm with an implication type** (e.g., Product t-norm + S-implication), enabling a principled investigation into how different logical priors affect learned representations. To ground this learning process, we rely on the following theorem.

**Theorem 1** (Consistency with Classical Semantics). *Let $\mathcal{O}$ be a fuzzy $\mathcal{ALC}$ ontology, $\varphi$ be a fuzzy assertion, and $\alpha \in (0.5, 1)$ be the truth threshold for the crisp transformation $\sharp(\cdot)$. Then:*

$$\mathcal{O} \approx \varphi \iff \sharp\mathcal{O} \models \sharp\varphi$$

*where $\approx$ denotes fuzzy entailment and $\models$ denotes classical three-valued entailment.*

#### 3.2.2 NORMALIZATION FOR A MODULAR LOSS DESIGN

Directly translating arbitrary $\mathcal{ALC}$ axioms into a loss function is intractable due to their nested and complex structure. To create a modular and manageable learning objective, we first simplify the ontology's syntax via a normalization process. The goal is to decompose every complex axiom into a set of semantically equivalent, simple axioms that conform to one of the seven elementary **Normal Forms** shown in Figure 1.

This decomposition is achieved in three steps: (i) applying the **Standard Transformations** from Figure 1 to push negations to atomic concepts (negation normal form); (ii) recursively applying the **Normalization Rules** from Figure 1 to break down complex concepts, introducing fresh concept names as placeholders; (iii) using a classical reasoner to interpret these new names. This entire process is guaranteed to be semantics-preserving.

**Theorem 2** (Semantic Preservation via Conservative Extension (Baader et al., 2017)). *For any $\mathcal{ALC}$ ontology $\mathcal{O}$, the normalization process produces $\mathcal{O}'$ in polynomial time such that: (i) every model of $\mathcal{O}'$ is a model of $\mathcal{O}$, and (ii) for every model $\mathcal{I}$ of $\mathcal{O}$ there exists a model $\mathcal{I}'$ of $\mathcal{O}'$ such that the interpretations of the concept and role names from sig($\mathcal{O}$) coincide in $\mathcal{I}$ and $\mathcal{I}'$.*

### 3.3 LOSS FUNCTION STRATEGIES

With a normalized TBox $\mathcal{T}'$ and a chosen set of fuzzy operators, we can now define our loss functions, which aim to minimize the dissatisfaction of the logical axioms.

| Normalization Rules | Normal Forms |
|---|---|
| NR1. $\hat{D} \sqsubseteq \hat{E} \Rightarrow \hat{D} \sqsubseteq A,\ A \sqsubseteq \hat{E}$ | NF1. $C \sqsubseteq B$ |
| NR2. $\hat{D} \sqcap C \sqsubseteq B \Rightarrow \hat{D} \sqsubseteq A,\ A \sqcap C \sqsubseteq B$ | NF2. $C_1 \sqcap C_2 \sqsubseteq B$ |
| NR3. $C \sqcup \hat{D} \sqsubseteq B \Leftrightarrow \hat{D} \sqsubseteq B,\ C \sqsubseteq B$ | NF3. $B \sqsubseteq C_1 \sqcup C_2$ |
| NR4. $\exists r.\hat{D} \sqsubseteq B \Rightarrow \hat{D} \sqsubseteq A,\ \exists r.A \sqsubseteq B$ | NF4. $C \sqsubseteq \exists r.B$ |
| NR5. $\forall r.\hat{D} \sqsubseteq B \Rightarrow \hat{D} \sqsubseteq A,\ \forall r.A \sqsubseteq B$ | NF5. $C \sqsubseteq \forall r.B$ |
| NR6. $B \sqsubseteq D \sqcap E \Leftrightarrow B \sqsubseteq D,\ B \sqsubseteq E$ | NF6. $\exists r.B \sqsubseteq C$ |
| NR7. $B \sqsubseteq D \sqcup \hat{E} \Rightarrow B \sqsubseteq D \sqcup A,\ A \sqsubseteq \hat{E}$ | NF7. $\forall r.B \sqsubseteq C$ |
| NR8. $B \sqsubseteq \exists r.\hat{D} \Rightarrow A \sqsubseteq \hat{D},\ B \sqsubseteq \exists r.A$ | **where:** |
| NR9. $B \sqsubseteq \forall r.\hat{D} \Rightarrow A \sqsubseteq \hat{D},\ B \sqsubseteq \forall r.A$ | – $\hat{D}, \hat{E}$: complex concepts (not $\top$, $\bot$, or atomic) |
| **Standard Transformations:** | – $A$: newly introduced concept name |
| $\neg(C \sqcap D) \Leftrightarrow \neg C \sqcup \neg D \quad \neg(C \sqcup D) \Leftrightarrow \neg C \sqcap \neg D$ | – $B$: concept name or its negation |
| $\neg \exists r.C \Leftrightarrow \forall r.\neg C \quad \neg \forall r.C \Leftrightarrow \exists r.\neg C \quad \neg\neg C \Leftrightarrow C$ | – $C, D, E$: arbitrary concepts |
| | – $r$: role name |

Figure 1: Normalization Rules and Normal Forms

### 3.3.1 HIERARCHICAL LOSS

The most direct objective is to enforce global consistency across all axioms. The hierarchical loss penalizes the violation of each subsumption axiom, averaged over all individuals:

$$\mathcal{L}_H(\mathcal{I}, \mathcal{T}') = \frac{1}{|\mathcal{T}'|} \sum_{(C \sqsubseteq D) \in \mathcal{T}'} \frac{1}{|\Delta|} \sum_{a \in \Delta} \left(1 - I\left(C^{\mathcal{I}}(a), D^{\mathcal{I}}(a)\right)\right)$$

where $I(u, v)$ represents the chosen fuzzy implication (S-, R-, or sigmoidal-implication). A fuzzy implication evaluates to 1 if and only if the subsumption $u \leq v$ is satisfied. Therefore, minimizing $1 - I$ drives the model towards an interpretation where all axioms hold. When this loss converges to 0, the learned interpretation becomes a valid model of the ontology.

### 3.3.2 RULE-BASED LOSS: A SEMANTICALLY-GATED APPROACH

While the hierarchical loss enforces global consistency, its learning signal can be diffuse. It informs the model that a violation has occurred but offers limited guidance on how to resolve it. This can lead to **representational collapse**, where the model finds trivial solutions (e.g., learning $C^{\mathcal{I}} = D^{\mathcal{I}} = 0.5$ for an axiom $C \sqsubseteq D$) that satisfy the loss without capturing the intended semantics.

We define representational collapse as the failure to learn any meaningful interpretation, manifesting either as *collapse to emptiness* (where concepts map to 0, vacuously satisfying axioms) or *collapse to ambiguity* (where concepts cluster around 0.5, representing maximum uncertainty). This is distinct from *reasoning shortcuts* (Bortolotti et al., 2024; Marconato et al., 2025), which typically refer to models achieving high confidence on incorrect semantic mappings.

To address this, we introduce a rule-based loss, namely $\mathcal{L}_R$, designed to provide a more targeted and semantically-aware learning signal. The core idea is to modulate the penalty for a violated axiom based on the model's confidence in its prediction. This is achieved by introducing a **confidence-based semantic gate**, which scales the loss according to the predicted truth value of the axiom's conclusion.

For any normalized axiom of the form $C \sqsubseteq D$, the rule-based loss is defined as:

$$\mathcal{L}_R(\mathcal{I}, C \sqsubseteq D) = \frac{1}{|\Delta|} \sum_{a \in \Delta} \underbrace{\left(1 - D^{\mathcal{I}}(a)\right)}_{\text{Semantic Gate}} \cdot \underbrace{\left(1 - I\left(C^{\mathcal{I}}(a), D^{\mathcal{I}}(a)\right)\right)}_{\text{Dissatisfaction Penalty}}$$

The total rule-based loss is the average over all axioms in $\mathcal{T}'$. For axioms involving quantifiers (NF4-NF7), $C$ and $D$ simply represent the complex concepts on the left and right sides of the subsumption, with their fuzzy interpretations defined in Section 2.2.

**Mitigating Representational Collapse** The semantic gate $(1 - D^{\mathcal{I}}(a))$ is the key to mitigating collapse. Its effect is twofold, namely (i) **Preserving Correct Predictions**: When the model is

confident that the conclusion $D$ is true for an individual $a$ (i.e., $D^{\mathcal{I}}(a) \to 1$), the gate $(1 - D^{\mathcal{I}}(a)) \to 0$. This effectively silences the loss for that individual, preventing the optimizer from needlessly altering already correct and high-confidence predictions; (ii) **Targeting High-Confidence Errors**: When the model is confidently wrong—believing the premise $C$ is true ($C^{\mathcal{I}}(a) \to 1$) while believing the conclusion $D$ is false ($D^{\mathcal{I}}(a) \to 0$)—the semantic gate approaches 1. This delivers the maximum possible penalty, forcing the optimizer to prioritize resolving the most flagrant logical contradictions.

This approach fundamentally alters the optimization landscape. It discourages the model from hiding in ambiguous, low-confidence states (e.g., $C^{\mathcal{I}}(a) = D^{\mathcal{I}}(a) = 0.1$) because a low value for $D^{\mathcal{I}}(a)$ creates a large gate, resulting in a significant penalty that pushes the model towards a more decisive and logically consistent state. While this mechanism effectively addresses optimization pathologies related to collapse, we acknowledge that true semantic correctness may require additional signals (e.g., concept supervision) beyond the scope of this unsupervised framework.

### 3.3.3 ADAPTIVE DUAL-LOSS

A fixed weighting between the hierarchical loss ($\mathcal{L}_H$) and the rule-based loss ($\mathcal{L}_R$) may be suboptimal. Ontologies rich in existential ($\exists$) or universal ($\forall$) quantifiers often require a stronger emphasis on $\mathcal{L}_R$ to accurately capture role semantics.

To address this, our third strategy is an adaptive dual-loss mechanism. Let $\omega_H$ and $\omega_R$ be the weights for $\mathcal{L}_H$ and $\mathcal{L}_R$ respectively, with $\omega_H + \omega_R = 1$. The weight $\omega_R$ is tied to the prevalence of quantified axioms within the normalized TBox $\mathcal{T}'$:

$$\omega_R = \sigma \left( \lambda \cdot \frac{|\{\text{axiom} \in \mathcal{T}' : \text{axiom is of form NF4-NF7}\}|}{|\mathcal{T}'|} \right)$$

where $\sigma(\cdot)$ is the sigmoid function, and $\lambda$ is a scaling hyperparameter. This formulation automatically increases the influence of $\mathcal{L}_R$ for ontologies with a higher proportion of quantified constraints. The final total loss is:

$$\mathcal{L}_{\text{total}} = (1 - \omega_R) \cdot \mathcal{L}_H + \omega_R \cdot \mathcal{L}_R$$

This creates a curriculum that effectively responds to the specific reasoning challenges presented by the ontology's structure. Unlike "rule weighting" schemes in prior works (Hu et al., 2016; Marra et al., 2019a) which assign confidence scores to individual logical formulas, our Adaptive Dual-Loss is a **structure-aware optimization mechanism**. It balances two distinct *optimization strategies* applied globally to the entire ontology, dynamically shifting focus to the gated loss strategy ($\mathcal{L}_R$) exactly when the ontological structure (i.e., high quantifier density) suggests a risk of gradient sparsity.

The soundness of these learning objectives is formally established by the following theorem. We premise the theorem on a standard property of fuzzy implications, which ensures a strict correspondence between the optimization objective and the logical goal. Specifically, we require the chosen implication operator $I(u, v)$ to be *separating*, meaning it satisfies the condition that $I(u, v) = 1 \iff u \leq v$. This property is crucial as it guarantees that a zero-loss state is achieved if and only if the underlying logical subsumption is fully satisfied. We verify that this condition holds for the primary operators used in our evaluation, particularly the R-implications derived from Gödel, Product, and Łukasiewicz t-norms, thus ensuring our theoretical guarantees are directly applicable to our empirical results.

**Theorem 3** (Soundness of the Learning Objectives). *Let $\mathcal{O}'$ be a normalized ontology and $\mathcal{I}$ be a learned interpretation. Assuming the chosen fuzzy implication $I(u, v)$ satisfies the property that $I(u, v) = 1 \iff u \leq v$, then all of our proposed loss objectives are sound: (i) If the Hierarchical Loss $\mathcal{L}_H(\mathcal{I}, \mathcal{T}') = 0$, then $\mathcal{I}$ is a valid fuzzy model of $\mathcal{O}'$; (ii) If the Rule-Based Loss $\mathcal{L}_R(\mathcal{I}, \mathcal{T}') = 0$, then $\mathcal{I}$ is a valid fuzzy model of $\mathcal{O}'$; (iii) Consequently, if the Adaptive Dual-Loss $\mathcal{L}_{total} = 0$, then $\mathcal{I}$ is a valid fuzzy model of $\mathcal{O}'$.*

## 4 EXPERIMENTS

Our empirical evaluation is designed to validate the central claims of our work. We aim to answer four key research questions (RQs): **(RQ1)** How does the choice of different fuzzy operators (t-norms and implications) affect learning performance across a variety of ontologies? **(RQ2)** How robust is

our framework at recovering knowledge from noisy and incomplete data compared to established baselines? **(RQ3)** Does our adaptive dual-loss strategy provide a tangible benefit over fixed-loss approaches? **(RQ4)** How does NeSy$\mathcal{ALC}$ perform on a practical, end-to-end neural-symbolic task that integrates perceptual input with logical reasoning?

We design three complementary experimental tasks. Task 1 (Masked ABox Revision) evaluates core symbolic reasoning capabilities, while Task 2 (Semantic Image Interpretation) validates end-to-end neural-symbolic integration with real perceptual data. Additional experiments on Conjunctive Query Answering (Task 3) are provided in the Appendix.

### 4.1 TASK 1: MASKED ABOX REVISION (MAR)

This task directly evaluates the core reasoning and learning capability of our framework (**RQ1, RQ2, RQ3**). The goal is to measure how effectively a model can use the logical constraints in a TBox to correct and complete a noisy, incomplete ABox. For a given ontology, we first use a classical reasoner HermiT (Glimm et al., 2014) to generate an **ideal grounding** $\mathcal{I}$ which serves as the ground truth. We then simulate a noisy perceptual input $\mathcal{I}'$ by randomly **masking** a portion of the assertions in $\mathcal{I}$, setting their truth values to the maximally uncertain range of $[0.2, 0.8]$. The model is initialized with this noisy grounding $\mathcal{I}'$ and must learn a revised grounding $\mathcal{I}''$ that satisfies the TBox. The quality of the revision is measured by comparing the final learned grounding $\mathcal{I}''$ against the ideal grounding $\mathcal{I}$. We use a **crisp transformation** (values $> 0.8$ are considered true, $< 0.2$ false) to compute the **Success Rate**, which is the percentage of TBox axioms satisfied by the final grounding.

#### 4.1.1 EXPERIMENTAL SETUP

We compare against **nine** established baselines spanning three paradigms:

- **Geometric Embedding Frameworks** ($\mathcal{EL}^{++}$): BoxEL (Zhang et al., 2020), Box$^2$EL (Jackermeier et al., 2024), and EL Embedding (Kulmanov et al., 2019).
- **Fuzzy Frameworks** (FOL and $\mathcal{ALC}$): LTN (Badreddine et al., 2022), F$\mathcal{ALC}$ON (Tang et al., 2022a), and CatE (Zhapa-Camacho & Hoehndorf, 2024).
- **Probabilistic Frameworks** (Probabilistic Logic (Nilsson, 1986)): DeepProbLog (Manhaeve et al., 2018), DeepStochLog (Winters et al., 2022), and NeurASP (Yang et al., 2020).

We evaluate on eight ontologies from BioPortal (Noy et al., 2008), selected to ensure diversity in scale, structure, and domain. The suite spans compact taxonomies (Sso: 176 TBox, 0 ABox) to large knowledge bases (Nifdys: 3188 TBox, 102 ABox), with quantifier density ranging from 0% to 42.27% (Ontodm). Domains cover social relationships (Family), clinical assessments (Nihss), and biochemical structures (Glycan). The selection deliberately includes challenging cases: ontologies with inconsistent ABox assertions (Ontodm, Nifdys) and varying knowledge completeness (Family vs. augmented Family2). Detailed statistics are provided in Appendix. For FOL-based baselines, we convert each $\mathcal{ALC}$ ontology into equivalent FOL formulas using the standard translation (Baader et al., 2017).

To investigate **RQ1**, our operator-agnostic framework was evaluated with a comprehensive suite of fuzzy semantics: **Gödel**, **Product**, **Łukasiewicz**, **Hamacher**, and **Yager** t-norms, paired with **S-implications**, **R-implications**, and **Sigmoidal implications**. All models were trained using the **Adam optimizer** with early stopping (patience: 10 epochs).

#### 4.1.2 RESULTS AND ANALYSIS

The results, shown in Table 1, validate our core claims. Regarding **RQ1**, the choice of fuzzy operators significantly impacts performance, with no single operator being universally dominant. Combinations using **R-implications** frequently outperform S-implication counterparts on ontologies with complex subsumption hierarchies. The **Sigmoidal implications** show competitive performance, achieving best or second-best results in 12 out of 144 dataset-configuration combinations.

Regarding **RQ2**, NeSy$\mathcal{ALC}$ consistently outperforms all baselines. While LTN and BoxEL perform well on specific ontologies (e.g., Ontodm), they lack consistency across the benchmark suite. The probabilistic baselines (DeepProbLog, DeepStochLog, NeurASP) achieve competitive performance on smaller ontologies, with DeepStochLog reaching 100% on Family. However, their performance

Table 1: Performance comparison on MAR Task with 20% mask rate across eight ontology datasets. Results show success rates (%) with standard deviations. Models are categorized into five groups: **Hierarchical Loss Models** (with S-/R-/Sigmoidal implications), **Rule-based Loss Models**, **Fixed Dual Loss Models** (equal weighting with $\omega_R = 0.5$), **Adaptive Dual-Loss Models** (dynamic $\omega_R$ based on quantifier prevalence), and **Baseline Models**. Best performance per dataset is shown in **bold**, second-best underlined. For Adaptive Dual-Loss, the cells in red indicate higher performance than Fixed Dual Loss, while green indicates lower performance.

| Model | Family | Family2 | Glycan | Glyco | Nifdys | Nihss | Ontodm | Sso |
|---|---|---|---|---|---|---|---|---|
| *Hierarchical Loss Models* | | | | | | | | |
| Gödel+S | 100.0±0.0 | 82.0±2.1 | 98.1±1.2 | 92.4±1.8 | 87.7±2.5 | 100.0±0.0 | 66.9±3.2 | 100.0±0.0 |
| Gödel+R | 93.1±1.9 | 96.7±1.3 | 86.0±2.8 | 92.9±2.0 | 95.0±1.7 | 100.0±0.0 | 65.2±3.1 | 100.0±0.0 |
| Gödel+Sig | 93.7±1.2 | 89.4±2.0 | 98.3±1.2 | 92.8±1.1 | 86.1±3.1 | 100.0±0.0 | 71.2±2.1 | 100.0±0.0 |
| Product+S | 100.0±0.0 | 80.3±2.4 | 98.1±1.2 | 94.9±1.5 | 94.9±1.6 | 100.0±0.0 | 80.2±2.8 | 100.0±0.0 |
| Product+R | 100.0±0.0 | 70.5±3.0 | 99.2±0.8 | 95.9±1.3 | 93.6±1.8 | 100.0±0.0 | 93.9±1.9 | 100.0±0.0 |
| Product+Sig | 98.6±0.6 | 71.6±3.2 | 97.4±0.8 | 95.2±2.3 | 93.7±1.1 | 100.0±0.0 | 79.7±2.5 | 100.0±0.0 |
| Łukasiewicz+S | 96.6±1.4 | 73.8±2.9 | 96.5±1.6 | 92.9±2.0 | 93.9±1.8 | 100.0±0.0 | 75.0±3.0 | 99.4±0.6 |
| Łukasiewicz+R | 100.0±0.0 | 78.7±2.6 | 99.2±0.8 | 97.0±1.1 | 94.1±1.7 | 100.0±0.0 | 96.3±1.2 | 100.0±0.0 |
| Łukasiewicz+Sig | 100.0±0.0 | 76.7±4.0 | 99.2±0.3 | 96.5±0.4 | 93.1±1.4 | 97.0±1.2 | 91.8±1.7 | 99.8±0.1 |
| Yager(p=2)+S | 100.0±0.0 | 78.7±2.8 | 96.5±1.2 | 92.9±1.1 | 89.6±2.9 | 100.0±0.0 | 80.4±2.8 | 100.0±0.0 |
| Yager(p=2)+R | 100.0±0.0 | 67.2±3.5 | 99.6±0.2 | 95.9±0.9 | 93.9±1.1 | 99.3±0.4 | 96.3±1.7 | 100.0±0.0 |
| Yager(p=2)+Sig | 100.0±0.0 | 76.5±3.4 | 98.8±0.4 | 93.4±1.5 | 94.1±0.6 | 100.0±0.0 | 85.3±2.0 | 100.0±0.0 |
| Hamacher+S | 93.1±1.9 | 59.0±3.5 | 98.1±1.2 | 94.9±1.5 | 90.1±2.3 | 99.1±0.9 | 79.4±2.9 | 100.0±0.0 |
| Hamacher+R | 93.1±1.9 | 96.7±1.3 | 100.0±0.0 | 95.9±1.3 | 98.5±0.9 | 100.0±0.0 | 85.0±2.7 | 99.1±0.9 |
| Hamacher+Sig | 92.7±1.8 | 68.9±4.1 | 98.5±0.5 | 94.2±1.6 | 92.7±1.3 | 99.4±0.4 | 80.2±4.1 | 100.0±0.0 |
| *Rule-based Loss Models* | | | | | | | | |
| Gödel+S | 100.0±0.0 | 100.0±0.0 | 100.0±0.0 | 91.9±2.1 | 94.2±1.8 | 100.0±0.0 | 91.2±2.2 | 100.0±0.0 |
| Gödel+R | 100.0±0.0 | 97.7±0.8 | 96.6±1.4 | 90.2±2.2 | 93.3±1.1 | 100.0±0.0 | 86.9±1.2 | 100.0±0.0 |
| Gödel+Sig | 100.0±0.0 | 96.0±1.6 | 98.7±0.4 | 92.1±1.9 | 94.3±1.2 | 100.0±0.0 | 92.0±1.5 | 100.0±0.0 |
| Product+S | 100.0±0.0 | 83.6±2.5 | 99.2±0.8 | 95.9±1.3 | 93.5±1.9 | 100.0±0.0 | 89.6±2.3 | 100.0±0.0 |
| Product+R | 100.0±0.0 | 89.5±1.4 | 98.7±0.9 | 94.5±1.4 | 94.1±1.3 | 99.6±0.2 | 90.2±2.1 | 100.0±0.0 |
| Product+Sig | 100.0±0.0 | 85.5±2.1 | 99.0±0.7 | 96.1±1.1 | 93.3±1.5 | 100.0±0.0 | 90.0±1.9 | 100.0±0.0 |
| Łukasiewicz+S | 100.0±0.0 | 73.8±2.9 | 70.9±3.4 | 60.9±3.7 | 97.4±1.0 | 80.6±3.2 | 86.5±2.6 | 94.2±1.7 |
| Łukasiewicz+R | 100.0±0.0 | 77.3±2.9 | 71.8±3.1 | 66.5±2.8 | 97.2±0.9 | 82.6±1.4 | 83.5±1.8 | 96.2±0.9 |
| Łukasiewicz+Sig | 100.0±0.0 | 69.1±3.3 | 67.9±3.2 | 64.2±3.1 | 97.3±1.0 | 76.6±3.5 | 92.3±2.3 | 97.4±0.6 |
| Yager(p=2)+S | 100.0±0.0 | 100.0±0.0 | 98.4±1.1 | 93.9±1.8 | 97.6±1.0 | 100.0±0.0 | 90.8±2.2 | 100.0±0.0 |
| Yager(p=2)+R | 100.0±0.0 | 99.4±0.3 | 98.7±0.3 | 92.5±1.8 | 95.5±1.1 | 100.0±0.0 | 91.3±1.3 | 100.0±0.0 |
| Yager(p=2)+Sig | 100.0±0.0 | 100.0±0.0 | 98.4±1.0 | 92.6±0.9 | 97.1±1.2 | 100.0±0.0 | 89.9±2.5 | 100.0±0.0 |
| Hamacher+S | 100.0±0.0 | 96.7±1.3 | 99.2±0.8 | 94.9±1.5 | 93.8±1.8 | 100.0±0.0 | 90.7±2.2 | 98.8±1.0 |
| Hamacher+R | 100.0±0.0 | 96.8±1.0 | 97.3±1.5 | 93.0±1.6 | 94.2±1.3 | 100.0±0.0 | 88.5±1.4 | 99.2±0.3 |
| Hamacher+Sig | 100.0±0.0 | 97.1±0.8 | 98.0±0.5 | 94.5±1.5 | 93.3±2.0 | 100.0±0.0 | 89.3±1.7 | 97.9±0.9 |
| *Fixed Dual Loss Models ($\omega_R = 0.5$)* | | | | | | | | |
| Gödel+S | 100.0±0.0 | 100.0±0.0 | 100.0±0.0 | 91.9±2.1 | 94.2±1.8 | 100.0±0.0 | 91.2±2.2 | 100.0±0.0 |
| Gödel+R | 100.0±0.0 | 97.7±0.8 | 96.6±1.4 | 90.2±2.2 | 93.3±1.1 | 100.0±0.0 | 86.9±1.2 | 100.0±0.0 |
| Gödel+Sig | 100.0±0.0 | 96.0±1.6 | 98.7±0.4 | 92.1±1.9 | 94.3±1.2 | 100.0±0.0 | 92.0±1.5 | 100.0±0.0 |
| Product+S | 100.0±0.0 | 83.6±2.5 | 99.2±0.8 | 95.9±1.3 | 93.5±1.9 | 100.0±0.0 | 89.6±2.3 | 100.0±0.0 |
| Product+R | 100.0±0.0 | 89.5±1.4 | 98.7±0.9 | 94.5±1.4 | 94.1±1.3 | 99.6±0.2 | 90.2±2.1 | 100.0±0.0 |
| Product+Sig | 100.0±0.0 | 85.5±2.1 | 99.0±0.7 | 96.1±1.1 | 93.3±1.5 | 100.0±0.0 | 90.0±1.9 | 100.0±0.0 |
| Łukasiewicz+S | 100.0±0.0 | 73.8±2.9 | 70.9±3.4 | 60.9±3.7 | 97.4±1.0 | 80.6±3.2 | 86.5±2.6 | 94.2±1.7 |
| Łukasiewicz+R | 100.0±0.0 | 77.3±2.9 | 71.8±3.1 | 66.5±2.8 | 97.2±0.9 | 82.6±1.4 | 83.5±1.8 | 96.2±0.9 |
| Łukasiewicz+Sig | 100.0±0.0 | 69.1±3.3 | 67.9±3.2 | 64.2±3.1 | 97.3±1.0 | 76.6±3.5 | 92.3±2.3 | 97.4±0.6 |
| Yager(p=2)+S | 100.0±0.0 | 85.2±2.4 | 100.0±0.0 | 97.0±0.9 | 93.0±1.8 | 100.0±0.0 | 92.9±1.0 | 100.0±0.0 |
| Yager(p=2)+R | 100.0±0.0 | 90.1±1.3 | 99.5±0.3 | 95.8±1.2 | 95.9±0.8 | 100.0±0.0 | 89.6±1.2 | 100.0±0.0 |
| Yager(p=2)+Sig | 100.0±0.0 | 93.5±1.7 | 100.0±0.0 | 99.0±0.3 | 96.4±0.7 | 100.0±0.0 | 88.8±1.4 | 100.0±0.0 |
| Hamacher+S | 93.3±1.7 | 89.3±2.6 | 99.1±0.5 | 94.2±1.2 | 93.2±1.8 | 96.5±0.9 | 81.6±2.4 | 98.7±0.1 |
| Hamacher+R | 97.9±1.1 | 78.6±3.2 | 95.9±1.3 | 94.1±1.2 | 94.7±1.3 | 90.5±1.4 | 83.7±2.6 | 100.0±0.0 |
| Hamacher+Sig | 90.8±1.3 | 90.6±1.5 | 97.6±0.5 | 91.7±1.0 | 90.6±1.8 | 91.3±1.7 | 87.2±2.7 | 99.4±0.3 |
| *Adaptive Dual-Loss Models* | | | | | | | | |
| Gödel+S | 100.0±0.0 | 96.9±1.2 | 98.6±1.0 | 94.6±1.6 | 94.8±1.6 | 100.0±0.0 | 87.3±2.5 | 100.0±0.0 |
| Gödel+R | 99.1±0.1 | 95.3±1.1 | 97.0±1.1 | 94.5±1.2 | 95.9±1.1 | 100.0±0.0 | 89.1±2.0 | 100.0±0.0 |
| Gödel+Sig | 100.0±0.0 | 97.0±1.2 | 98.1±0.8 | 93.2±1.9 | 93.9±1.3 | 100.0±0.0 | 90.1±1.7 | 100.0±0.0 |
| Product+S | 100.0±0.0 | 88.5±2.3 | 95.7±1.5 | 96.2±1.2 | 95.2±1.5 | 100.0±0.0 | 93.2±1.9 | 100.0±0.0 |
| Product+R | 100.0±0.0 | 89.6±2.0 | 94.3±1.8 | 97.1±1.3 | 91.9±1.2 | 100.0±0.0 | 92.4±1.2 | 100.0±0.0 |
| Product+Sig | 100.0±0.0 | 91.1±1.6 | 93.4±1.0 | 95.6±1.7 | 93.4±1.0 | 100.0±0.0 | 93.1±1.5 | 100.0±0.0 |
| Łukasiewicz+S | 100.0±0.0 | 89.6±2.2 | 99.2±0.8 | 92.2±2.0 | 97.9±1.0 | 92.1±2.1 | 88.7±2.4 | 98.2±1.1 |
| Łukasiewicz+R | 100.0±0.0 | 91.4±1.8 | 99.4±0.4 | 90.8±1.4 | 96.3±1.2 | 94.3±2.2 | 91.0±1.9 | 97.6±1.2 |
| Łukasiewicz+Sig | 99.5±0.4 | 90.0±1.4 | 98.2±1.1 | 93.2±1.8 | 97.9±1.0 | 92.4±2.5 | 85.7±2.7 | 99.1±0.4 |
| Yager(p=2)+S | 100.0±0.0 | 96.7±1.4 | 100.0±0.0 | 99.0±0.2 | 97.9±1.2 | 100.0±0.0 | 91.9±1.2 | 100.0±0.0 |
| Yager(p=2)+R | 100.0±0.0 | 95.6±1.7 | 99.8±0.1 | 99.0±0.2 | 96.4±1.3 | 100.0±0.0 | 92.4±1.1 | 100.0±0.0 |
| Yager(p=2)+Sig | 100.0±0.0 | 93.9±1.6 | 100.0±0.0 | 98.6±0.7 | 98.0±0.7 | 100.0±0.0 | 90.5±1.1 | 100.0±0.0 |
| Hamacher+S | 94.2±1.6 | 88.3±2.6 | 98.3±1.1 | 94.2±1.2 | 96.5±1.2 | 93.3±1.3 | 82.5±3.1 | 99.8±0.1 |
| Hamacher+R | 97.8±1.1 | 84.2±2.9 | 97.5±1.5 | 93.5±1.1 | 97.1±1.3 | 92.6±1.1 | 80.4±2.4 | 100.0±0.0 |
| Hamacher+Sig | 92.4±1.3 | 90.1±1.9 | 98.0±0.8 | 93.1±1.5 | 95.2±1.3 | 94.1±1.0 | 81.6±2.1 | 99.5±0.2 |
| *Baseline Models* | | | | | | | | |
| LTN (FOL) | 100.0±0.0 | 91.8±2.1 | 96.8±1.4 | 74.6±3.2 | 97.2±1.0 | 48.4±3.8 | 98.0±1.1 | 93.6±1.8 |
| $\mathcal{FALCON}$ ($\mathcal{ALC}$) | 100.0±0.0 | 78.7±2.6 | 86.3±2.7 | 64.5±3.6 | 94.9±1.6 | 48.4±3.8 | 88.6±2.4 | 93.6±1.8 |
| CatE ($\mathcal{ALC}$) | 100.0±0.0 | 90.5±2.1 | 95.8±1.9 | 77.2±2.0 | 97.6±0.5 | 85.1±3.1 | 97.8±0.5 | 97.0±0.9 |
| BoxEL ($\mathcal{EL}^{++}$) | 93.1±1.9 | 91.8±2.1 | 91.6±2.2 | 63.5±3.6 | 97.2±1.0 | 80.6±3.2 | 98.2±1.1 | 92.1±2.0 |
| Box$^2$EL ($\mathcal{EL}^{++}$) | 100.0±0.0 | 78.7±2.6 | 95.3±1.6 | 64.5±3.6 | 95.1±1.5 | 86.7±2.8 | 88.5±2.4 | 96.7±1.3 |
| ELEmbedding ($\mathcal{EL}^{++}$) | 100.0±0.0 | 80.8±2.4 | 90.2±2.3 | 75.1±3.1 | 99.8±0.2 | 58.1±3.5 | 95.8±1.4 | 97.1±1.2 |
| DeepProbLog (Prob.) | 99.1±0.2 | 86.5±3.4 | 95.1±1.8 | 75.1±2.8 | 95.8±1.7 | 83.8±2.2 | 98.0±0.7 | 94.5±1.6 |
| DeepStochLog (Prob.) | 100.0±0.0 | 91.2±2.7 | 96.2±1.3 | 70.9±3.5 | 94.1±1.2 | 88.9±2.5 | 97.4±1.2 | 96.8±1.2 |
| NeurASP (Prob.) | 100.0±0.0 | 90.6±2.5 | 96.9±1.8 | 78.1±2.5 | 92.7±1.3 | 75.1±2.8 | 95.7±1.3 | 94.5±1.6 |

degrades on ontologies with higher quantifier density and complex role restrictions (e.g., 70.9% on Family2 vs. our 96.7%), likely due to combinatorial grounding costs and difficulty encoding DL's open-domain quantification within probabilistic frameworks.

**Statistical Significance.** Welch's t-tests with Holm-Bonferroni correction comparing our best model against the best baseline yield statistically significant improvements ($p < 0.01$) on 5 of 8 datasets: Family2 ($p < 0.001$), Glycan ($p = 0.002$), Glyco ($p < 0.001$), Nihss ($p < 0.001$), and Sso ($p = 0.003$). The remaining datasets exhibit ceiling effects (Family, Nifdys at >99%) or competitive baseline performance (Ontodm, where BoxEL achieves 98.2%).

**Empirical Guidance for Operator Selection.** Our systematic study reveals actionable patterns:

- **T-norm Selection.** For *deeply nested axioms*, Product and Hamacher t-norms enable gradient penetration through nested quantifiers via strict monotonicity. Gödel's single-path gradients risk sparsity but remain effective with Rule-Based loss, where the confidence-aware weighting in $\mathcal{L}_R$ compensates (Gödel+S: 100% on Family, Family2, and Glycan in Rule-Based configurations).
- **Implication Selection.** For *structure learning tasks*, R-implications excel due to the separating property $\mathcal{I}(a, b) = 1 \Leftrightarrow a \leq b$, which creates a "logical hinge" producing gradients only when constraints are violated (Hamacher+R: 96.7% vs. Hamacher+S: 59.0% on Family2). For *perceptual integration tasks*, S-implications and Sigmoidal implications provide more symmetric gradient flow, as validated in the Semantic Image Interpretation experiments (Section 4.2).

These patterns provide principled starting points; however, ontology-specific characteristics (quantifier density, role complexity) significantly influence optimal choices.

## 4.2 Task 2: Semantic Image Interpretation (SII)

This task validates NeSy$\mathcal{ALC}$'s capability for end-to-end neural-symbolic integration, demonstrating that our framework can ground symbolic $\mathcal{ALC}$ ontological knowledge directly into visual features extracted from real images (**RQ4**). Unlike NeSy approaches that add logical constraints as regularization to supervised learning, NeSy$\mathcal{ALC}$ treats the logical knowledge base itself as the primary learning objective—the goal is to learn fuzzy interpretations $(C^{\mathcal{I}}, r^{\mathcal{I}})$ that satisfy the axioms, rather than to improve a separate neural task with logical guidance.

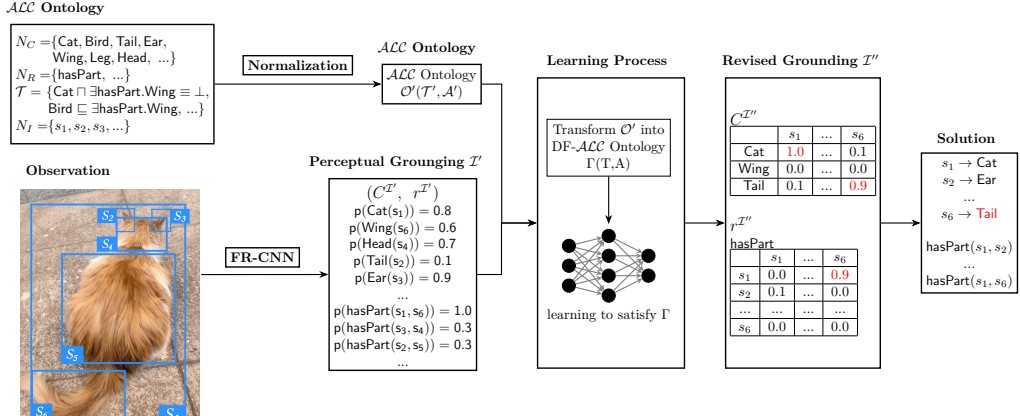

Figure 2: Semantic Image Interpretation workflow: Fast R-CNN provides noisy perceptual grounding, which NeSy$\mathcal{ALC}$ refines using ontological constraints.

### 4.2.1 Experimental Setup

We employ the **PASCAL-PART dataset** (Chen et al., 2014), a widely adopted benchmark from LTN's original SII experiments (Badreddine et al., 2022). A pre-trained, frozen **Fast R-CNN** (Girshick, 2015) object detector provides initial, noisy perceptual grounding $\mathcal{I}'$ of objects, parts, and spatial relations. NeSy$\mathcal{ALC}$ then refines $\mathcal{I}'$ using domain ontologies that encodes common-sense constraints.

Domain ontologies were manually constructed using **Protégé** (Musen, 2015), the de-facto industry-standard tool for ontology engineering. They contain $\mathcal{ALC}$ axioms that force the model to perform genuine multi-hop logical reasoning over noisy perceptual inputs, including: **Disjointness constraints:** e.g., Chair $\sqcap$ Table $\sqsubseteq$ $\perp$, **Part-whole reasoning:** e.g., Wheel $\sqsubseteq$ $\exists$partOf.Vehicle, and **Cardinality and spatial relations:** reasoning about object counts and positions under occlusion.

The complete OWL ontology files are provided in the supplementary material. We compare against the original **Fast R-CNN** output and **LTN** (Badreddine et al., 2022), measuring object classification performance (Precision, Recall, F1-score) after logical refinement.

Our goal in SII is *logical refinement of perception*, not joint perceptual-logical training. The detector remains frozen; it provides noisy grounding, and NeSy$\mathcal{ALC}$ learns interpretations that satisfy the ontology using $\mathcal{L}_{\text{total}}$ alone, with no additional perceptual loss term. This demonstrates the framework's versatility: it can work both (i) purely from logical knowledge (Task 1: MAR), and (ii) to refine noisy perceptual groundings (Task 2: SII), a flexibility that perception-dependent approaches lack.

### 4.2.2 RESULTS AND ANALYSIS

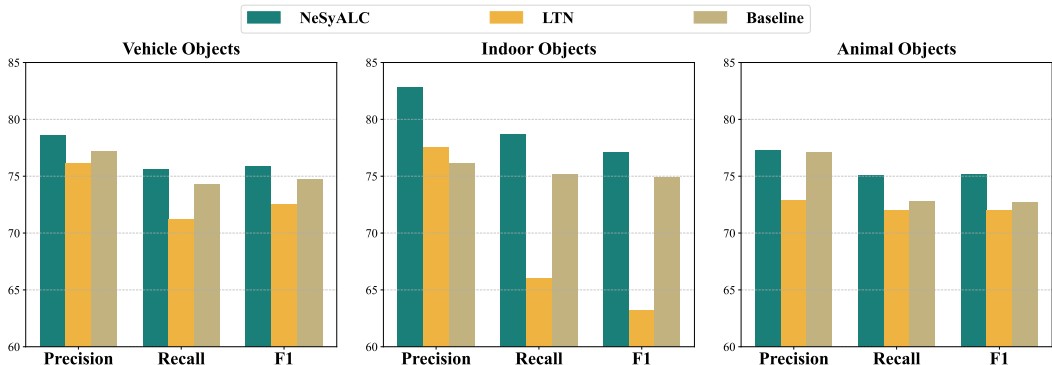

Figure 3: Object classification performance (macro-averaged) on PASCAL-PART across vehicle, indoor, and animal categories. NeSy$\mathcal{ALC}$ consistently outperforms the base **Fast R-CNN** detector and LTN across all metrics.

As shown in Figure 3, NeSy$\mathcal{ALC}$ significantly improves object classification by leveraging ontological constraints to refine noisy detector outputs. The framework achieves substantial F1 gains: **+7.8%** for indoor objects and **+5.2%** for animals. These improvements stem from the ontology's ability to resolve ambiguities that confuse purely perceptual systems (e.g., distinguishing chairs from tables based on part-whole relations rather than visual similarity alone).

This experiment demonstrates that NeSy$\mathcal{ALC}$ provides an effective, *unsupervised* method for improving symbol grounding: no labeled examples are required beyond the ontology itself. The framework successfully integrates perceptual data with symbolic knowledge, validating its applicability to real-world neural-symbolic pipelines where logical refinement of noisy neural outputs is essential.

## 5 CONCLUSION AND FUTURE WORK

This work bridges a critical methodological gap in Neural-Symbolic AI by operationalizing fuzzy $\mathcal{ALC}$ for practical, gradient-based learning at scale. We introduced NeSy$\mathcal{ALC}$, an end-to-end operator-agnostic framework for fuzzy $\mathcal{ALC}$, and through a systematic empirical analysis, demonstrated that the choice of fuzzy semantics is a crucial, non-trivial determinant of learning outcomes. Our key contribution is a novel adaptive dual-loss strategy that tailors the learning objective to the ontology's logical structure, yielding significant and consistent performance gains over established baselines.

For future work, the impact of operator choice motivates the development of methods to automatically learn the optimal fuzzy semantics for a given knowledge base, treating logical priors themselves as learnable components. Another promising opportunity lies in integrating NeSy$\mathcal{ALC}$ with LLMs, where it could serve as a powerful verifier to enforce logical consistency and mitigate hallucinations.

## ETHICS STATEMENT

This work adheres to the ICLR Code of Ethics. Our research is foundational in nature, focusing on computational principles of differentiable fuzzy Description Logic learning and reasoning. All datasets employed are publicly available ontology benchmarks; they contain no personally identifiable information and do not involve human subjects.

## REPRODUCIBILITY STATEMENT

We are committed to ensuring full reproducibility of our results. Complete implementation details, including network architectures, hyperparameter configurations, and training protocols, are provided in Appendix E. Source code and trained models, alongside all ontology datasets used in our experiments are included in the supplementary materials; we provide detailed statistics and preprocessing procedures in Appendix D.1.

## USE OF GENERATIVE AI TOOLS

The authors acknowledge the use of generative AI tools for light editing and refinement of human-authored text. All content, including research conception, methodology design, implementation, experimental execution, data analysis, and interpretation of findings, represents the original intellectual contribution of the human authors. No content was generated entirely by AI, and generative AI played no role in the scientific contributions of this work. The authors assume full responsibility for all content and claims presented herein.

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

# APPENDIX

## A   PROOFS OF THEORETICAL RESULTS

This appendix provides the formal proofs for the theorems presented in the main paper.

### A.1   PROOF OF THEOREM 1 (CONSISTENCY WITH CLASSICAL SEMANTICS)

**Theorem 1** (Consistency with Classical Semantics). *Let $\mathcal{O}$ be a fuzzy $\mathcal{ALC}$ ontology, $\varphi$ be a fuzzy assertion, and $\alpha \in (0.5, 1)$ be the truth threshold for the crisp transformation $\sharp(\cdot)$. Then:*

$$\mathcal{O} \approx \varphi \iff \sharp\mathcal{O} \models \sharp\varphi$$

*where $\approx$ denotes fuzzy entailment and $\models$ denotes classical three-valued entailment.*

*Proof.* The proof establishes the equivalence between fuzzy entailment ($\approx$) and entailment in a three-valued logic ($\models$) derived via a crisp transformation. Let the crisp transformation $\sharp(\cdot)$ for a fuzzy assertion $\varphi$ with a satisfaction degree of $n$ and a given threshold $\alpha \in (0.5, 1)$ be defined as:

$$\sharp\varphi = \begin{cases} \phi & \text{if } n > \alpha \\ \text{unknown} & \text{if } 1 - \alpha \le n \le \alpha \\ \neg\phi & \text{if } n < 1 - \alpha \end{cases}$$

We prove the biconditional statement ($\iff$) in two parts.

**Forward Direction ($\Rightarrow$): Assume $\mathcal{O} \approx \varphi$, we must show $\sharp\mathcal{O} \models \sharp\varphi$.**   The premise $\mathcal{O} \approx \varphi$ means that every fuzzy model of the ontology $\mathcal{O}$ also satisfies the fuzzy assertion $\varphi$. We need to show that any three-valued model of $\sharp\mathcal{O}$ must also be a model of $\sharp\varphi$.

Let $\mathcal{J}$ be an arbitrary three-valued model of $\sharp\mathcal{O}$. We construct a fuzzy interpretation $\mathcal{I}$ from $\mathcal{J}$ by canonically mapping the three-valued logic truth values to the fuzzy interval $[0, 1]$:

$$C^{\mathcal{I}}(a) := \begin{cases} 1 & \text{if } \mathcal{J} \models a : C \\ 0 & \text{if } \mathcal{J} \models a : \neg C \\ 0.5 & \text{if } a : C \text{ is unknown in } \mathcal{J} \end{cases}$$

By this construction, if $\mathcal{J}$ satisfies the constraints in $\sharp\mathcal{O}$, then $\mathcal{I}$ will satisfy the corresponding fuzzy constraints in $\mathcal{O}$. Thus, $\mathcal{I}$ is a valid fuzzy model of $\mathcal{O}$.

According to our initial assumption ($\mathcal{O} \approx \varphi$), since $\mathcal{I}$ is a model of $\mathcal{O}$, it must also satisfy $\varphi$. Applying the crisp transformation $\sharp(\cdot)$ back to the satisfaction degree of $\varphi$ in $\mathcal{I}$ must yield a result consistent with the three-valued logic of $\mathcal{J}$. Therefore, $\mathcal{J}$ must satisfy $\sharp\varphi$. As $\mathcal{J}$ was an arbitrary model of $\sharp\mathcal{O}$, we conclude that $\sharp\mathcal{O} \models \sharp\varphi$.

**Backward Direction ($\Leftarrow$): Assume $\sharp\mathcal{O} \models \sharp\varphi$, we must show $\mathcal{O} \approx \varphi$.**   The premise $\sharp\mathcal{O} \models \sharp\varphi$ means that every three-valued model of $\sharp\mathcal{O}$ also satisfies $\sharp\varphi$. We need to show that any fuzzy model of $\mathcal{O}$ must also satisfy $\varphi$.

Let $\mathcal{I}$ be an arbitrary fuzzy model of $\mathcal{O}$. We construct a three-valued interpretation $\mathcal{J}$ from $\mathcal{I}$ by applying the crisp transformation $\sharp(\cdot)$ to the membership degree of every assertion satisfied by $\mathcal{I}$. By this construction, if $\mathcal{I}$ is a fuzzy model of $\mathcal{O}$, then $\mathcal{J}$ must be a three-valued model of $\sharp\mathcal{O}$.

Given our initial assumption ($\sharp\mathcal{O} \models \sharp\varphi$), since $\mathcal{J}$ is a model of $\sharp\mathcal{O}$, it must also satisfy $\sharp\varphi$. By the definition of the transformation, if the transformed interpretation $\mathcal{J}$ satisfies $\sharp\varphi$, it implies that the original fuzzy interpretation $\mathcal{I}$ must have satisfied the fuzzy assertion $\varphi$.

Since this holds for any arbitrary fuzzy model $\mathcal{I}$ of $\mathcal{O}$, we conclude that $\mathcal{O} \approx \varphi$. $\qquad\square$

### A.2   PROOF OF THEOREM 2 (SEMANTIC PRESERVATION VIA CONSERVATIVE EXTENSION)

**Theorem 2** (Semantic Preservation via Conservative Extension (Baader et al., 2017)). *For any $\mathcal{ALC}$ ontology $\mathcal{O}$, the normalization process produces $\mathcal{O}'$ in polynomial time such that: (i) every model*

*of $\mathcal{O}'$ is a model of $\mathcal{O}$, and (ii) for every model $\mathcal{I}$ of $\mathcal{O}$ there exists a model $\mathcal{I}'$ of $\mathcal{O}'$ such that the interpretations of the concept and role names from sig($\mathcal{O}$) coincide in $\mathcal{I}$ and $\mathcal{I}'$.*

*Proof.* The proof adapts the standard approach for conservative extensions in DLs to our fuzzy semantics context (Baader et al., 2017). The property of being a conservative extension is transitive; therefore, it suffices to show that a single application of any normalization rule preserves the fuzzy model relationship. We provide a detailed proof for rule NR2 as a representative example.

**Rule NR2:** $C \sqcap \hat{D} \sqsubseteq B \Rightarrow \hat{D} \sqsubseteq A_{new}, C \sqcap A_{new} \sqsubseteq B$, where $A_{new} \in N_C$ is a fresh concept name not in sig($\mathcal{O}$). Let $\otimes$ be the chosen t-norm for conjunction.

**Part 1: Every fuzzy model of $\mathcal{O}'$ is a fuzzy model of $\mathcal{O}$.**

Let $\mathcal{I}$ be any fuzzy model of $\mathcal{O}'$. By definition, it satisfies the new axioms, meaning for any individual $a \in \Delta^{\mathcal{I}}$:

- $(\hat{D})^{\mathcal{I}}(a) \leq (A_{new})^{\mathcal{I}}(a)$
- $(C \sqcap A_{new})^{\mathcal{I}}(a) \leq B^{\mathcal{I}}(a)$, which means $C^{\mathcal{I}}(a) \otimes (A_{new})^{\mathcal{I}}(a) \leq B^{\mathcal{I}}(a)$

From (1) and the monotonicity of t-norms, we have $C^{\mathcal{I}}(a) \otimes (\hat{D})^{\mathcal{I}}(a) \leq C^{\mathcal{I}}(a) \otimes (A_{new})^{\mathcal{I}}(a)$. Combining this with (2), we get:

$$(C \sqcap \hat{D})^{\mathcal{I}}(a) = C^{\mathcal{I}}(a) \otimes (\hat{D})^{\mathcal{I}}(a) \leq C^{\mathcal{I}}(a) \otimes (A_{new})^{\mathcal{I}}(a) \leq B^{\mathcal{I}}(a)$$

This shows that $\mathcal{I}$ satisfies the original axiom $C \sqcap \hat{D} \sqsubseteq B$. Since all other axioms are unchanged, $\mathcal{I}$ is also a fuzzy model of $\mathcal{O}$.

**Part 2: For every fuzzy model of $\mathcal{O}$, there exists a fuzzy model of $\mathcal{O}'$ that agrees on sig($\mathcal{O}$).**

Let $\mathcal{I}$ be a fuzzy model of $\mathcal{O}$. We construct a new interpretation $\mathcal{I}'$ such that:

- For all symbols in sig($\mathcal{O}$), $\mathcal{I}'$ agrees with $\mathcal{I}$.
- For the new concept name $A_{new}$, we define its interpretation as $(A_{new})^{\mathcal{I}'}(a) := (\hat{D})^{\mathcal{I}}(a)$ for all $a \in \Delta^{\mathcal{I}}$.

We verify that $\mathcal{I}'$ satisfies the new axioms in $\mathcal{O}'$. For any $a \in \Delta^{\mathcal{I}}$:

- $\hat{D} \sqsubseteq A_{new}$: This requires $(\hat{D})^{\mathcal{I}'}(a) \leq (A_{new})^{\mathcal{I}'}(a)$. By our construction, this is $(\hat{D})^{\mathcal{I}}(a) \leq (\hat{D})^{\mathcal{I}}(a)$, which is true.
- $C \sqcap A_{new} \sqsubseteq B$: This requires $C^{\mathcal{I}'}(a) \otimes (A_{new})^{\mathcal{I}'}(a) \leq B^{\mathcal{I}'}(a)$. By construction, this becomes $C^{\mathcal{I}}(a) \otimes (\hat{D})^{\mathcal{I}}(a) \leq B^{\mathcal{I}}(a)$, which is true because $\mathcal{I}$ is a model of $\mathcal{O}$ and thus satisfies the original axiom.

Therefore, $\mathcal{I}'$ is a fuzzy model of $\mathcal{O}'$. The proofs for other rules follow analogously. The polynomial time complexity follows from the fact that each rule application reduces the nesting depth of axioms and adds at most a constant number of new symbols, ensuring the process terminates in polynomial time with respect to the size of the original ontology $\mathcal{O}$. $\square$

A.3 PROOF OF THEOREM 3 (SOUNDNESS OF LEARNING OBJECTIVES)

**Theorem 3** (Soundness of the Learning Objectives). *Let $\mathcal{O}'$ be a normalized ontology and $\mathcal{I}$ be a learned interpretation. Assuming the chosen fuzzy implication $I(u, v)$ satisfies the property that $I(u, v) = 1 \iff u \leq v$, then all of our proposed loss objectives are sound: (i) If the Hierarchical Loss $\mathcal{L}_H(\mathcal{I}, \mathcal{T}') = 0$, then $\mathcal{I}$ is a valid fuzzy model of $\mathcal{O}'$; (ii) If the Rule-Based Loss $\mathcal{L}_R(\mathcal{I}, \mathcal{T}') = 0$, then $\mathcal{I}$ is a valid fuzzy model of $\mathcal{O}'$; (iii) Consequently, if the Adaptive Dual-Loss $\mathcal{L}_{total} = 0$, then $\mathcal{I}$ is a valid fuzzy model of $\mathcal{O}'$.*

*Proof.* The proof demonstrates that for each of the three proposed loss objectives, convergence to a value of zero is a sufficient condition to guarantee that the learned interpretation $\mathcal{I}$ is a valid fuzzy model of the normalized ontology $\mathcal{O}'$. An interpretation $\mathcal{I}$ is considered a valid fuzzy model if, for every axiom $(C \sqsubseteq D) \in \mathcal{T}'$ and every individual $a \in \Delta$, the condition $C^{\mathcal{I}}(a) \leq D^{\mathcal{I}}(a)$ is satisfied. The proof relies on the theorem's central assumption that the chosen fuzzy implication $I(u, v)$ has the property that $I(u, v) = 1 \iff u \leq v$.

**Proof of (i): Soundness of Hierarchical Loss**  The Hierarchical Loss is defined as $\mathcal{L}_H = \frac{1}{|\mathcal{T}'||\Delta|} \sum_{(C \sqsubseteq D) \in \mathcal{T}'} \sum_{a \in \Delta} (1 - I(C^{\mathcal{I}}(a), D^{\mathcal{I}}(a)))$. Since fuzzy truth values and the outputs of fuzzy implications are in the range $[0, 1]$, each term $(1 - I(\cdot, \cdot))$ in the summation is non-negative. For the total sum $\mathcal{L}_H$ to be 0, every individual term must be 0. This implies that for all axioms $(C \sqsubseteq D)$ and all individuals $a$:

$$1 - I(C^{\mathcal{I}}(a), D^{\mathcal{I}}(a)) = 0 \implies I(C^{\mathcal{I}}(a), D^{\mathcal{I}}(a)) = 1$$

By the central assumption of the theorem, $I(u, v) = 1$ is equivalent to $u \leq v$. Therefore, for all axioms and individuals, $C^{\mathcal{I}}(a) \leq D^{\mathcal{I}}(a)$. This is the definition of a valid fuzzy model.

**Proof of (ii): Soundness of Rule-Based Loss**  The Rule-Based Loss is an average over terms of the form $(1 - D^{\mathcal{I}}(a)) \cdot (1 - I(C^{\mathcal{I}}(a), D^{\mathcal{I}}(a)))$. All components of this product are non-negative. For the total loss $\mathcal{L}_R$ to be 0, each of these product terms must be 0 for all axioms and individuals. A product is zero if and only if at least one of its factors is zero. This means for each axiom and each individual $a$, one of the following two cases must hold:

- **Case A:** $1 - D^{\mathcal{I}}(a) = 0 \implies D^{\mathcal{I}}(a) = 1$. Since $C^{\mathcal{I}}(a)$ is a fuzzy membership degree in $[0, 1]$, the condition $C^{\mathcal{I}}(a) \leq 1$ is always true. Thus, the subsumption $C^{\mathcal{I}}(a) \leq D^{\mathcal{I}}(a)$ is satisfied.
- **Case B:** $1 - I(C^{\mathcal{I}}(a), D^{\mathcal{I}}(a)) = 0 \implies I(C^{\mathcal{I}}(a), D^{\mathcal{I}}(a)) = 1$. By the central assumption, this is equivalent to $C^{\mathcal{I}}(a) \leq D^{\mathcal{I}}(a)$. The subsumption is satisfied.

In both possible cases, the logical condition $C^{\mathcal{I}}(a) \leq D^{\mathcal{I}}(a)$ is satisfied. Since this holds for all axioms, if $\mathcal{L}_R = 0$, then $\mathcal{I}$ is a valid fuzzy model.

**Proof of (iii): Soundness of Adaptive Dual-Loss**  The total loss is defined as $\mathcal{L}_{\text{total}} = (1 - \omega_R) \cdot \mathcal{L}_H + \omega_R \cdot \mathcal{L}_R$. The weights $\omega_R$ and $(1 - \omega_R)$ are in the range $[0, 1]$. As established in the proofs for (i) and (ii), both $\mathcal{L}_H$ and $\mathcal{L}_R$ are non-negative. The total loss is a weighted sum of non-negative values, which is itself non-negative. The only way for $\mathcal{L}_{\text{total}}$ to be 0 is if both terms in the sum are 0. This implies that $\mathcal{L}_H = 0$ and $\mathcal{L}_R = 0$. From the proofs of (i) and (ii), this is a sufficient condition for $\mathcal{I}$ to be a valid fuzzy model of $\mathcal{O}'$. $\qquad\square$

# B  COMPUTATIONAL COMPLEXITY ANALYSIS

NeSy$\mathcal{ALC}$ involves three main computational phases: ontology normalization, fuzzy interpretation construction, and dual-loss optimization. We analyze the complexity of each phase systematically to demonstrate the practical feasibility of our approach.

**Notation.** For an $\mathcal{ALC}$ ontology $\mathcal{O}$, we use:

- $|N_I|$: number of individual names in the domain ($|\Delta| = |N_I|$).
- $|N_C|$: number of concept names.
- $|N_R|$: number of role names.
- $|\mathcal{T}'|$: number of axioms after normalization.
- $d$: maximum nesting depth of concepts.

**Theorem 4** (Normalization Complexity). *Given an $\mathcal{ALC}$ ontology $\mathcal{O}$ with $|\mathcal{T}|$ TBox axioms, the normalization process runs in $O(|\mathcal{T}| \cdot d)$ time, where $d$ is the maximum depth of concept nesting in any axiom.*

*Proof.* Each axiom is processed once. For an axiom with maximum nesting depth $d$, decomposition requires at most $d$ recursive steps. Each step takes constant time. The total number of generated axioms $|\mathcal{T}'|$ is bounded by $O(|\mathcal{T}| \cdot d)$. Computing interpretations for new concept names requires polynomial time with a classical reasoner. $\qquad\square$

**Theorem 5** (Space Complexity). *The space requirements for storing fuzzy interpretations are $O(|N_C| \cdot |N_I| + |N_R| \cdot |N_I|^2)$.*

*Proof.* Each of the $|N_C|$ concepts requires a vector of size $|N_I|$. Each of the $|N_R|$ roles requires a matrix of size $|N_I| \times |N_I|$. The total space is the sum of these components. $\qquad\square$

**Theorem 6** (Time Complexity per Training Iteration). *Each training iteration of the dual-loss optimization requires $O(|\mathcal{T}'| \cdot |N_I|^2)$ time in the worst case.*

*Proof.* The complexity is dominated by the forward pass loss calculation. We analyze the cost for computing the loss contribution of a single axiom over all individuals. Accessing any $C^{\mathcal{I}}(a)$ or $r^{\mathcal{I}}(a, b)$ is an $O(1)$ operation.

**Case 1: Propositional Axioms (NF1–NF3).** These axioms (e.g., $C_1 \sqcap C_2 \sqsubseteq B$) involve only Boolean connectives. For a single individual $a$, calculating the membership degree requires a constant number of $O(1)$ operations. Thus, iterating over all $|N_I|$ individuals takes $O(|N_I|)$ time per axiom.

**Case 2: Quantified Axioms (NF4–NF7).** These axioms (e.g., $\exists r.B \sqsubseteq C$) are the computational bottleneck. To find the membership degree $(\exists r.B)^{\mathcal{I}}(a)$, one must compute $\sup_{b \in \Delta} \{r^{\mathcal{I}}(a, b) \otimes B^{\mathcal{I}}(b)\}$. For a **single individual** $a$, this requires iterating through all $|N_I|$ individuals $b$, performing $O(1)$ work for each, and then finding the supremum over $|N_I|$ values. This takes $O(|N_I|)$ time. To compute the total loss for this axiom, this $O(|N_I|)$ operation must be repeated for each of the $|N_I|$ individuals, resulting in a total complexity of $O(|N_I|^2)$ per quantified axiom.

**Total Complexity.** The total loss is the sum over all $|\mathcal{T}'|$ axioms. In the worst case, all axioms are quantified. Therefore, the total complexity is $O(|\mathcal{T}'| \cdot |N_I|^2)$. As established in Theorem 2, $|\mathcal{T}'|$ is polynomially bounded by the size of the original ontology, ensuring the overall process is polynomial. $\square$

**Practical Scalability.** The quadratic dependence on $|N_I|$ is a worst-case theoretical bound. In practice, this is mitigated by several factors:

- **Sparsity:** Real-world role matrices are typically highly sparse (>90%).
- **Batch Processing:** Training on subsets of individuals reduces the effective $|N_I|$ per step.
- **Early Convergence:** Logical constraints provide strong guidance, accelerating convergence and reducing the required number of epochs.

These factors enable NeSy$\mathcal{ALC}$ to scale effectively to real-world ontologies.

## C    RELATED WORK

Neural-symbolic AI seeks to unify the complementary strengths of neural learning and symbolic reasoning (d'Avila Garcez et al., 2002; Besold et al., 2021). The primary challenge is the *representational incompatibility* between continuous neural representations and discrete symbolic structures (Harnad, 1990), a modern incarnation of the symbol grounding problem (Harnad, 1990). The field's evolution can be understood as a progression through different paradigms for bridging this divide.

### C.1    FOUNDATIONS: THE LOGICAL DICHOTOMY IN NEURAL-SYMBOLIC AI

At the heart of NeSy systems lies a foundational choice of logical formalism, which typically falls into one of two categories, creating a dichotomy between expressivity and computational tractability.

**High-Expressivity Logics.** A major branch of research leverages the rich syntax of full FOL to model complex relations. These systems, often relaxed with fuzzy or probabilistic semantics, provide powerful tools for reasoning over learned representations. This category includes a diverse set of influential frameworks such as Logic Tensor Networks (LTN) (Badreddine et al., 2022) and its variants like logLTN (Badreddine et al., 2023), NTP (Rocktäschel & Riedel, 2017), TensorLog (Cohen et al., 2020), and other semantic entailment systems like F$\mathcal{ALC}$ON (Tang et al., 2022a) and TAR (Tang et al., 2022b). While offering maximal expressive power, they inherit FOL's theoretical undecidability, which poses challenges for scalability and formal guarantees in complex scenarios.

**Tractable, Scalable Logics.** In contrast, to ensure computational feasibility, another significant branch focuses on decidable fragments of logic, primarily the $\mathcal{EL}$ family of DLs. These systems utilize elegant geometric embeddings to capture logical semantics, where concepts are represented as regions in a vector space. This includes methods like EL Embeddings (Kulmanov et al., 2019), EmEL++ (Peng et al., 2022), and a series of box-based embedding models such as BoxEL (Xiong et al.,

2022) and Box$^2$EL (Jackermeier et al., 2024). The strength of these approaches is their polynomial-time reasoning, making them highly scalable. Their fundamental weakness, however, is severely limited expressivity, as they cannot represent negation, disjunction, or universal quantification.

## C.2 Methodologies for Differentiable Reasoning

Given a logical formalism, various methodologies have been developed to make reasoning differentiable and integrate it with neural networks.

**Modular and Probabilistic Integration.** One major paradigm maintains a degree of separation between neural and symbolic modules. This includes foundational systems like KBANN (Towell & Shavlik, 1994), as well as a rich lineage of modern probabilistic logic programming frameworks like ProbLog (Raedt et al., 2007), DeepProbLog (Manhaeve et al., 2018), NeurASP (Yang et al., 2020), and DeepStochLog (Winters et al., 2022). These systems excel at handling uncertainty in a principled way but often face challenges in deep, end-to-end optimization of the entire perception-reasoning pipeline.

**Logic as a Regularizer.** A widely adopted approach uses logic as a soft constraint during training. Here, logical axioms are converted into penalty terms in the loss function to guide the learning process towards consistent representations. This diverse category includes early work like MLN (Richardson & Domingos, 2006) and a variety of modern techniques such as Semantic-Based Regularization (SBR) (Diligenti et al., 2017), SL (Xu et al., 2018), DLM (Marra et al., 2019b), and SATNet (Wang et al., 2019), which integrates a differentiable SAT solver for propositional logic (Wang et al., 2019).

## C.3 Positioning Our Work

Our work, NeSy$\mathcal{ALC}$, is designed to carve a new path that addresses the limitations of the aforementioned approaches. We argue that the next frontier for NeSy is not just about choosing a point on the expressivity-tractability spectrum, but about building **robust and adaptive learning methodologies** for expressive logics.

- **On Formalism:** We choose $\mathcal{ALC}$ as our logical foundation. It occupies a crucial middle ground, offering the rich Boolean expressivity and quantifiers missing from the $\mathcal{EL}$ family, while remaining provably decidable, unlike FOL.
- **On Semantics:** We address a critical, underexplored methodological gap in the field. Most fuzzy-logic-based systems (e.g., (Badreddine et al., 2022)) commit to a single, hard-coded choice of fuzzy operators. This overlooks the fact that the optimal choice of semantics is highly task- and data-dependent (van Krieken et al., 2022). Our primary contribution is an **operator-agnostic learning framework**, the first of its kind for an expressive DL, which transforms the choice of fuzzy semantics from a fixed assumption into a subject of systematic inquiry.
- **On Learning:** The insights gained from our flexible framework directly motivate our second key contribution: a novel **adaptive dual-loss strategy**. This mechanism allows the learning objective to dynamically adjust to the logical structure of the knowledge base, enhancing learning robustness and moving beyond static, heuristic loss formulations.

In summary, while prior work has focused on establishing different logical foundations, NeSy$\mathcal{ALC}$ provides the practical learning machinery required to make an expressive yet decidable logic like $\mathcal{ALC}$ truly effective and robust for modern machine learning tasks.

# D Additional Experiments

This section provides a comprehensive expansion of the experimental results presented in the main paper. Our goal is to offer a deeper, more detailed analysis that complements the primary findings. Here, we present: (i) the complete performance metrics for the MAR task across all masking rates (40%, 60%, and 80%), allowing for a thorough assessment of model robustness under increasing uncertainty; and (ii) a detailed analysis of the Conjunctive Query Answering (CQA) task, including performance on ontologies with deliberately introduced imperfections.

## D.1 Benchmark Dataset Details

All experiments are conducted on a diverse suite of eight ontologies selected from BioPortal (Noy et al., 2008). Our selection was guided by a principled set of criteria aimed at ensuring a rigorous and multi-faceted evaluation, rather than choosing benchmarks that might narrowly favor our approach. Specifically, we considered three key dimensions: logical diversity, domain coverage, and robustness to real-world imperfections.

**Logical diversity.** The chosen ontologies span a wide range of $\mathcal{ALC}$ constructors and axiom patterns. At one end, ontologies such as `Sso` represent relatively simple taxonomic hierarchies with limited role usage. At the other end, `Family2` and `GlycoRDF` feature complex role restrictions, nested concept definitions, and substantial ABox assertions. This spectrum allows us to evaluate how our framework scales from lightweight terminologies to expressive knowledge bases.

**Domain coverage.** To mitigate domain-specific biases, we selected ontologies from a variety of domain areas: social and kinship relationships (`Family`, `Family2`), clinical assessments (`Nihss`), biochemical structures (`Glycan`, `GlycoRDF`), and scientific data management (`Ontodm`, `Nifdys`). This diversity ensures that our findings are not artifacts of a particular modeling style or vocabulary.

**Real-world imperfections.** Real-world ontologies often contain noise, inconsistencies, or incomplete information. To test our framework's resilience to such challenges, we deliberately included ontologies with known issues: `Ontodm` and `Nifdys` contain **inconsistent ABox assertions**, while comparing `Family` with the augmented `Family2` allows us to assess performance under **varying degrees of knowledge completeness**. These cases stress-test the framework beyond ideal settings.

Table 2: Detailed statistics of the eight benchmark ontologies from BioPortal, ordered by TBox size

| | Nihss (smallest) | Glyco | Family | Sso | Family2 | Glycan | Ontodm | Nifdys (largest) |
|---|---|---|---|---|---|---|---|---|
| *Scale* | | | | | | | | |
| # TBox Axioms | 318 | 1,453 | 2,032 | 2,050 | 2,054 | 2,422 | 3,476 | 6,435 |
| # ABox Axioms | 146 | 518 | 224 | 366 | 224 | 593 | 1,113 | 2,920 |
| # Concepts | 18 | 113 | 19 | 176 | 19 | 137 | 838 | 2,751 |
| # Roles | 16 | 91 | 4 | 22 | 4 | 109 | 78 | 68 |
| # Individuals | 106 | 219 | 202 | 158 | 202 | 247 | 187 | 102 |
| *Logical Structure* | | | | | | | | |
| Quantified Axioms (%) | 0.00 | 0.59 | 0.00 | 0.00 | 0.31 | 0.51 | **42.27** | 6.66 |
| Expressivity | ¬ | ¬, ⊥, ∃ | ¬ | — | ¬, ⊓, ∃ | ¬, ⊓, ∃ | ¬, ⊓, ⊥, ∃, ⊔ | ¬, ⊥, ∃ |
| *Domain* | | | | | | | | |
| Application Area | Clinical | Biochem. | Social | Social | Social | Biochem. | Data Mining | Neuralsci. |

Table 2 summarizes their key statistics, including size (number of TBox/ABox axioms, concepts, roles, and individuals) and logical expressivity (the presence of logical constructors like negation, conjunction, etc.). This diversity in structure and complexity allows for a rigorous evaluation of our framework's performance and robustness across a wide range of reasoning challenges.

## D.2 MAR: Operator Choice and Robustness Across All Masking Rates

While the main submission presents the MAR results for a 20% mask rate to establish a baseline, this section provides the complete results for significantly higher levels of uncertainty: 40% (Table 3), 60% (Table 4), and 80% (Table 5). These extended results facilitate a more rigorous evaluation of our framework's robustness. As the masking rate increases, the task becomes substantially more challenging, testing the limits of each model's capability to reason from sparse data. Across these tables, we observe a consistent trend: while the performance of all models naturally degrades with more missing information, our NeSy$\mathcal{ALC}$ variants, particularly those employing R-implications and our **Adaptive Dual-Loss** strategy, maintain a more pronounced performance advantage over the baselines. This demonstrates their superior capability to reconstruct knowledge and enforce logical consistency even when initialized with highly incomplete and uncertain information.

Table 3: Performance comparison on MAR Task with 40% mask rate across eight ontology datasets. Results show success rates (%) with standard deviations. Models are categorized into five groups: **Hierarchical Loss Models** (with S-/R-/Sigmoidal implications), **Rule-based Loss Models**, **Adaptive Dual-Loss Models** (dynamic $\omega_R$ based on quantifier prevalence), and **Baseline Models**. Best performance per dataset is shown in **bold**, second-best underlined.

| Mask Rate / Model | Family | Family2 | Glycan | Glyco | Nifdys | Nihss | Ontodm | Sso |
|---|---|---|---|---|---|---|---|---|
| **Mask Rate 0.4** | | | | | | | | |
| ***Hierarchy Models*** | | | | | | | | |
| Gödel+S | 89.7±2.3 | 77.0±2.8 | 95.0±1.7 | 89.3±2.4 | 84.2±2.9 | **100.0±0.0** | 61.7±3.8 | 99.4±0.6 |
| Gödel+R | 93.1±2.0 | **96.7±1.3** | 86.0±2.8 | 92.9±2.0 | 94.0±1.6 | **100.0±0.0** | 62.7±3.7 | **100.0±0.0** |
| Gödel+Sig | 88.3±2.9 | 84.4±3.6 | 95.1±1.5 | 89.8±2.1 | 83.1±3.0 | **100.0±0.0** | 67.3±3.1 | **100.0±0.0** |
| Product+S | **100.0±0.0** | 73.8±2.9 | 95.0±1.7 | 89.8±2.3 | 91.0±2.2 | **100.0±0.0** | 79.5±3.0 | 99.3±0.7 |
| Product+R | **100.0±0.0** | 75.2±2.7 | 99.2±0.8 | **95.9±1.2** | 93.5±1.8 | **100.0±0.0** | 87.9±2.5 | **100.0±0.0** |
| Product+Sig | **100.0±0.0** | 76.7±2.1 | 96.3±1.4 | 93.5±1.8 | 91.0±2.3 | **100.0±0.0** | 81.5±3.3 | **100.0±0.0** |
| Łukasiewicz+S | **100.0±0.0** | 70.5±3.2 | 95.7±1.5 | 89.3±2.4 | 88.3±2.6 | **100.0±0.0** | 67.3±3.5 | 99.1±0.9 |
| Łukasiewicz+R | **100.0±0.0** | 73.8±2.9 | 98.8±1.0 | **97.0±1.1** | 94.3±1.7 | **100.0±0.0** | **94.6±1.7** | **100.0±0.0** |
| Łukasiewicz+Sig | **100.0±0.0** | 67.3±3.1 | 96.2±1.5 | 93.5±2.0 | 92.9±2.4 | 98.1±0.7 | 83.1±2.5 | 99.4±0.4 |
| Yager(p=0.5)+S | 92.4±1.3 | 64.5±3.2 | 84.7±2.3 | 88.3±1.6 | 90.5±1.8 | **100.0±0.0** | 75.5±3.9 | 99.5±0.2 |
| Yager(p=0.5)+R | 91.5±1.2 | 66.3±3.9 | 85.4±2.5 | 90.5±1.4 | 98.1±0.5 | **100.0±0.0** | 94.1±1.5 | 99.1±0.3 |
| Yager(p=0.5)+Sig | 91.9±1.8 | 68.5±3.2 | 87.4±2.3 | 88.7±2.3 | 90.8±2.0 | 99.7±0.2 | 76.8±3.9 | **100.0±0.0** |
| Yager(p=2)+S | **100.0±0.0** | 77.1±2.9 | 93.8±1.6 | 93.1±1.4 | 87.5±2.8 | **100.0±0.0** | 80.8±2.5 | **100.0±0.0** |
| Yager(p=2)+R | **100.0±0.0** | 69.4±3.6 | 98.2±0.5 | 93.4±1.3 | 91.0±1.8 | 99.2±0.4 | 91.7±1.6 | **100.0±0.0** |
| Yager(p=2)+Sig | **100.0±0.0** | 73.5±3.2 | 95.0±0.8 | 92.9±1.6 | 94.2±1.4 | **100.0±0.0** | 83.6±2.8 | **100.0±0.0** |
| Hamacher+S | 93.1±2.0 | 37.7±4.5 | 96.1±1.4 | 94.1±1.8 | 85.8±2.8 | **100.0±0.0** | 76.2±3.1 | **100.0±0.0** |
| Hamacher+R | 93.3±1.9 | **96.7±1.3** | **100.0±0.0** | 95.9±1.3 | 97.2±1.0 | **100.0±0.0** | 82.1±2.7 | **100.0±0.0** |
| Hamacher+Sig | 92.8±1.2 | 72.9±3.0 | 96.2±1.7 | 95.4±1.6 | 85.3±2.9 | **100.0±0.0** | 80.9±2.2 | **100.0±0.0** |
| ***Rule-Based Models*** | | | | | | | | |
| Gödel+S | **100.0±0.0** | **100.0±0.0** | 98.2±1.2 | 90.9±2.2 | 94.1±1.8 | **100.0±0.0** | 90.9±2.3 | **100.0±0.0** |
| Gödel+R | **100.0±0.0** | 95.6±1.7 | 95.1±2.0 | 90.0±1.8 | 92.7±1.3 | **100.0±0.0** | 86.1±2.4 | **100.0±0.0** |
| Gödel+Sig | **100.0±0.0** | 94.8±2.1 | 96.4±1.7 | 91.7±2.1 | 94.1±1.5 | **100.0±0.0** | 91.6±2.0 | **100.0±0.0** |
| Product+S | **100.0±0.0** | 80.3±2.7 | 99.1±0.9 | 93.9±1.9 | 93.6±1.8 | **100.0±0.0** | 87.5±2.6 | **100.0±0.0** |
| Product+R | **100.0±0.0** | 88.1±1.9 | 97.3±1.1 | 90.0±1.8 | 93.9±1.3 | 99.5±0.3 | 88.6±1.2 | **100.0±0.0** |
| Product+Sig | **100.0±0.0** | 83.7±2.5 | 99.5±0.3 | 92.5±1.7 | 91.1±2.0 | **100.0±0.0** | 85.9±2.5 | **100.0±0.0** |
| Łukasiewicz+S | **100.0±0.0** | 72.1±3.1 | 63.2±3.9 | 54.3±4.2 | 88.5±2.6 | 68.7±3.8 | 86.2±2.8 | 93.6±2.0 |
| Łukasiewicz+R | **100.0±0.0** | 74.3±2.6 | 68.9±3.0 | 55.5±3.9 | 91.7±2.1 | 80.5±3.0 | 81.0±3.2 | 92.4±3.6 |
| Łukasiewicz+Sig | **100.0±0.0** | 73.5±2.5 | 61.3±4.1 | 61.9±5.2 | 91.5±2.0 | 69.7±4.2 | 89.9±2.3 | 93.5±2.0 |
| Yager(p=0.5)+S | 93.1±2.0 | 80.2±2.7 | 48.4±4.3 | 47.7±4.4 | 98.5±1.0 | 48.9±4.4 | 92.4±2.1 | 93.2±2.0 |
| Yager(p=0.5)+R | 98.6±1.1 | 83.1±2.2 | 58.3±4.7 | 51.6±4.5 | 97.4±1.4 | 56.1±3.5 | 91.2±2.0 | 92.6±2.8 |
| Yager(p=0.5)+Sig | 96.2±1.4 | 80.0±2.9 | 57.1±3.6 | 49.6±3.5 | 98.0±0.8 | 61.2±2.3 | 90.4±2.1 | 92.4±1.6 |
| Yager(p=2)+S | **100.0±0.0** | **100.0±0.0** | 98.4±1.1 | 93.4±1.9 | 97.5±1.2 | **100.0±0.0** | 87.0±2.7 | **100.0±0.0** |
| Yager(p=2)+R | **100.0±0.0** | 99.4±0.4 | 98.6±0.5 | 91.3±2.0 | 94.6±1.2 | **100.0±0.0** | 89.4±1.9 | **100.0±0.0** |
| Yager(p=2)+Sig | **100.0±0.0** | **100.0±0.0** | 94.7±1.7 | 93.5±1.9 | 95.1±1.5 | **100.0±0.0** | 90.4±2.0 | **100.0±0.0** |
| Hamacher+S | **100.0±0.0** | 96.7±1.3 | 99.2±0.8 | 94.4±1.7 | 93.7±1.8 | **100.0±0.0** | 90.4±2.2 | 98.8±1.0 |
| Hamacher+R | **100.0±0.0** | 96.9±1.0 | 98.7±0.6 | 92.7±1.8 | 93.9±1.5 | **100.0±0.0** | 89.4±1.9 | 99.1±0.5 |
| Hamacher+Sig | **100.0±0.0** | 96.0±1.4 | 97.4±1.5 | 93.8±1.7 | 92.9±1.3 | **100.0±0.0** | 87.5±2.1 | 96.2±1.2 |
| ***Adaptive Dual-Loss*** | | | | | | | | |
| Gödel+S | **100.0±0.0** | 85.9±2.5 | **99.3±0.2** | 78.2±3.2 | 96.8±1.3 | **100.0±0.0** | 81.2±3.0 | 96.7±1.4 |
| Gödel+R | 98.6±1.1 | 94.8±1.3 | 96.7±1.0 | 93.2±1.5 | 94.8±1.4 | **100.0±0.0** | 86.7±2.5 | **100.0±0.0** |
| Gödel+Sig | **100.0±0.0** | 97.0±1.2 | 97.8±0.9 | 92.4±1.5 | 91.5±1.3 | **100.0±0.0** | 89.3±2.0 | **100.0±0.0** |
| Product+S | 96.1±1.4 | **99.0±0.9** | **99.6±0.1** | 85.6±2.7 | 86.1±2.8 | 99.7±0.3 | **95.8±1.5** | 98.8±1.0 |
| Product+R | **100.0±0.0** | 92.3±1.4 | 93.1±1.5 | 87.4±1.7 | 86.2±2.1 | 99.2±0.4 | 94.5±1.4 | **100.0±0.0** |
| Product+Sig | **100.0±0.0** | 90.9±1.5 | 92.8±1.1 | 85.6±2.7 | 86.3±2.4 | 98.5±0.8 | 85.9±2.7 | 99.5±0.2 |
| Łukasiewicz+S | 94.7±1.6 | 79.6±2.9 | 76.4±3.4 | **96.7±1.4** | 92.1±2.2 | 87.6±2.6 | 84.3±2.9 | 93.9±1.9 |
| Łukasiewicz+R | **100.0±0.0** | 83.4±3.0 | 95.3±2.2 | 91.7±2.1 | 91.8±2.1 | 88.9±1.8 | 90.7±2.0 | 94.6±1.3 |
| Łukasiewicz+Sig | **100.0±0.0** | 85.7±2.3 | 94.2±1.5 | 90.8±2.9 | 90.2±1.8 | 84.3±2.5 | 86.7±2.2 | 95.7±1.6 |
| Yager(p=0.5)+S | 84.7±2.8 | 72.4±3.1 | 86.2±4.7 | 91.7±2.0 | 90.8±2.3 | 98.7±0.4 | 78.1±3.8 | **100.0±0.0** |
| Yager(p=0.5)+R | 88.8±1.2 | 74.6±2.7 | 85.4±2.0 | 90.6±1.3 | 91.5±1.9 | 99.4±0.3 | 80.4±2.0 | **100.0±0.0** |
| Yager(p=0.5)+Sig | 83.6±2.0 | 72.5±3.0 | 83.2±2.8 | 91.5±2.4 | 89.5±1.7 | 98.5±0.7 | 78.5±3.0 | **100.0±0.0** |
| Yager(p=2)+S | **100.0±0.0** | 94.7±1.4 | 99.1±0.4 | 97.8±1.1 | 95.8±1.2 | **100.0±0.0** | 89.7±1.3 | **100.0±0.0** |
| Yager(p=2)+R | **100.0±0.0** | 92.6±1.1 | 98.9±0.5 | 94.8±1.4 | 94.9±1.5 | **100.0±0.0** | 86.5±2.0 | **100.0±0.0** |
| Yager(p=2)+Sig | **100.0±0.0** | 95.0±1.2 | 98.4±0.7 | 95.7±1.2 | 95.5±1.6 | **100.0±0.0** | 89.5±1.9 | **100.0±0.0** |
| Hamacher+S | 98.9±1.0 | 98.2±1.2 | 95.8±1.0 | 94.8±1.6 | **97.4±1.2** | 99.5±0.5 | 92.7±2.1 | 99.3±0.2 |
| Hamacher+R | 93.1±2.7 | 86.5±2.1 | 96.7±1.2 | 92.8±1.9 | 96.2±1.4 | 91.6±2.4 | 84.6±2.3 | 98.3±1.5 |
| Hamacher+Sig | 91.7±1.9 | 88.6±2.0 | 95.8±1.0 | 91.5±1.5 | 94.3±1.4 | 93.6±1.7 | 86.4±1.8 | 99.1±0.4 |
| ***Baselines*** | | | | | | | | |
| LTN | **100.0±0.0** | 91.8±2.1 | 91.5±2.3 | 74.1±3.6 | 96.1±1.4 | 48.2±4.6 | **98.1±1.2** | 93.6±1.9 |
| F$\mathcal{ALCON}$ | **100.0±0.0** | 78.1±3.0 | 85.5±2.8 | 64.1±3.8 | 93.2±2.0 | 48.4±4.7 | 97.6±1.3 | 93.2±1.9 |
| BoxEL | 92.9±2.2 | 91.8±2.1 | 91.1±2.4 | 62.9±3.9 | 96.2±1.4 | 48.4±4.0 | 86.2±2.9 | 93.6±1.9 |
| Box2EL | **100.0±0.0** | 78.5±2.9 | 94.7±1.8 | 60.4±4.0 | 95.1±1.5 | 53.9±3.8 | 86.4±2.8 | 93.2±1.9 |
| ELEmbedding | **100.0±0.0** | 80.3±2.8 | 89.8±2.5 | 68.5±3.7 | **99.7±0.1** | 51.6±4.3 | 95.1±1.7 | 97.2±1.1 |
| CatE | **100.0±0.0** | 88.6±2.5 | 91.3±1.3 | 75.8±2.3 | 94.9±1.1 | 82.5±2.9 | 96.3±1.0 | 95.4±1.4 |
| DeepProbLog | 97.7±0.7 | 83.2±3.5 | 93.1±1.8 | 72.9±2.3 | 93.2±1.6 | 82.4±2.5 | 96.3±1.2 | 91.7±1.8 |
| DeepStochLog | **100.0±0.0** | 88.5±2.3 | 93.2±1.7 | 73.0±2.7 | 91.9±2.1 | **87.4±2.2** | 95.9±1.4 | 95.1±1.5 |
| NeurASP | **100.0±0.0** | 89.2±2.3 | 95.8±1.3 | **76.9±2.1** | 89.6±2.5 | 74.3±1.9 | 93.1±1.9 | 91.6±1.2 |

Table 4: Performance comparison on MAR Task with 60% mask rate across eight ontology datasets. Results show success rates (%) with standard deviations. Models are categorized into five groups: **Hierarchical Loss Models** (with S-/R-/Sigmoidal implications), **Rule-based Loss Models**, **Adaptive Dual-Loss Models** (dynamic $\omega_R$ based on quantifier prevalence), and **Baseline Models**. Best performance per dataset is shown in **bold**, second-best underlined.

| Mask Rate / Model | Family | Family2 | Glycan | Glyco | Nifdys | Nihss | Ontodm | Sso |
|---|---|---|---|---|---|---|---|---|
| **Mask Rate 0.6** | | | | | | | | |
| ***Hierarchy Models*** | | | | | | | | |
| Gödel+S | 79.3±1.5 | 63.9±2.8 | 95.3±1.7 | 89.2±2.4 | 78.3±2.9 | **100.0±0.0** | 60.1±3.8 | 99.4±0.1 |
| Gödel+R | 92.1±1.8 | 95.2±2.4 | 85.1±2.9 | 90.2±1.6 | 93.8±1.8 | **100.0±0.0** | 62.6±4.1 | **100.0±0.0** |
| Gödel+Sig | 84.5±2.6 | 80.2±3.1 | 92.0±1.9 | 87.1±2.8 | 80.2±3.0 | **100.0±0.0** | 64.0±3.5 | **100.0±0.0** |
| Product+S | 96.6±2.1 | 73.2±1.6 | 93.0±1.5 | 89.8±2.3 | 89.0±2.5 | **100.0±0.0** | 77.0±2.8 | **100.0±0.0** |
| Product+R | **100.0±0.0** | 75.1±2.8 | 98.9±1.1 | **95.9±0.3** | 93.5±1.1 | **100.0±0.0** | 85.0±2.2 | **100.0±0.0** |
| Product+Sig | 98.5±1.4 | 73.0±2.5 | 94.6±1.6 | 91.4±2.1 | 88.0±2.8 | **100.0±0.0** | 79.2±3.0 | **100.0±0.0** |
| Łukasiewicz+S | 79.7±2.1 | 70.5±2.4 | 93.4±0.8 | 89.1±2.0 | 88.3±2.6 | **100.0±0.0** | 65.9±1.5 | **100.0±0.0** |
| Łukasiewicz+R | **100.0±0.0** | 73.2±2.1 | 98.8±0.3 | **96.4±1.7** | 93.9±1.7 | **100.0±0.0** | 94.4±1.2 | **100.0±0.0** |
| Łukasiewicz+Sig | 100.0±0.0 | 65.0±3.3 | 94.0±1.8 | 91.0±2.5 | 90.0±2.6 | 98.1±0.7 | 80.5±2.8 | 99.0±0.5 |
| Yager(p=0.5)+S | 90.0±1.8 | 61.8±3.5 | 82.0±3.0 | 86.5±2.6 | 88.0±2.4 | **100.0±0.0** | 73.0±3.9 | 98.8±0.6 |
| Yager(p=0.5)+R | 89.0±1.9 | 64.0±3.2 | 83.0±3.2 | 88.0±2.1 | 95.5±1.4 | **100.0±0.0** | 92.0±1.8 | 98.5±0.7 |
| Yager(p=0.5)+Sig | 90.0±1.7 | 66.0±3.0 | 85.0±2.8 | 86.5±2.9 | 88.5±2.1 | 99.7±0.2 | 74.5±3.6 | **100.0±0.0** |
| Yager(p=2)+S | **100.0±0.0** | 74.0±2.6 | 91.0±1.7 | 90.0±2.0 | 85.0±2.8 | **100.0±0.0** | 78.0±2.5 | **100.0±0.0** |
| Yager(p=2)+R | **100.0±0.0** | 67.0±3.1 | 96.0±0.9 | 91.5±1.6 | 89.5±1.9 | 99.2±0.4 | 90.0±1.9 | **100.0±0.0** |
| Yager(p=2)+Sig | **100.0±0.0** | 71.0±2.8 | 93.0±1.1 | 91.0±1.8 | 92.0±1.6 | **100.0±0.0** | 82.0±2.6 | **100.0±0.0** |
| Hamacher+S | 85.5±4.3 | 41.0±4.9 | 96.0±1.2 | 93.9±1.5 | 82.1±3.8 | **100.0±0.0** | 74.5±3.3 | **100.0±0.0** |
| Hamacher+R | 93.1±1.7 | 96.5±0.8 | **100.0±0.0** | 95.4±1.7 | 95.9±1.6 | **100.0±0.0** | 77.5±2.2 | **100.0±0.0** |
| Hamacher+Sig | 90.5±2.1 | 70.0±3.0 | 94.0±1.5 | 94.0±1.6 | 83.0±3.0 | **100.0±0.0** | 79.0±2.6 | **100.0±0.0** |
| ***Rule-Based Models*** | | | | | | | | |
| Gödel+S | **100.0±0.0** | **100.0±0.0** | 96.8±1.5 | 89.2±2.4 | 92.7±2.1 | **100.0±0.0** | 88.6±2.7 | **100.0±0.0** |
| Gödel+R | 98.5±1.2 | 94.0±2.1 | 93.5±2.3 | 89.0±2.2 | 91.0±1.9 | 99.6±0.4 | 85.0±2.6 | 99.5±0.5 |
| Gödel+Sig | 99.0±0.8 | 92.5±1.9 | 95.0±1.6 | 90.0±1.8 | 92.5±1.7 | 99.9±0.2 | 90.0±2.0 | **100.0±0.0** |
| Product+S | **100.0±0.0** | 78.9±3.1 | 97.4±1.2 | 91.7±2.2 | 91.8±2.3 | **100.0±0.0** | 85.2±3.0 | **100.0±0.0** |
| Product+R | 99.0±1.0 | 86.0±2.2 | 95.5±1.4 | 89.0±2.0 | 92.0±1.8 | 99.0±0.5 | 86.5±1.9 | **100.0±0.0** |
| Product+Sig | 99.0±1.0 | 81.0±2.5 | 97.5±1.1 | 90.0±2.1 | 89.0±2.0 | 99.6±0.4 | 84.0±2.6 | 99.5±0.3 |
| Łukasiewicz+S | **100.0±0.0** | 70.3±3.4 | 61.8±4.1 | 52.1±4.5 | 86.9±2.9 | 66.4±4.0 | 84.7±3.2 | 91.8±2.3 |
| Łukasiewicz+R | 100.0±0.0 | 72.0±3.1 | 66.0±3.5 | 53.0±4.1 | 89.0±2.2 | 79.0±3.2 | 79.0±3.0 | 91.0±3.1 |
| Łukasiewicz+Sig | 100.0±0.0 | 71.0±2.9 | 59.0±4.0 | 59.0±4.7 | 89.0±2.1 | 68.0±3.6 | 88.0±2.8 | 92.0±2.0 |
| Yager(p=0.5)+S | 91.6±2.3 | 78.7±3.0 | 46.9±4.6 | 45.3±4.7 | **96.8±1.3** | 47.2±4.8 | 90.1±2.5 | 91.5±2.4 |
| Yager(p=0.5)+R | 96.0±1.6 | 80.0±2.4 | 55.0±4.8 | 49.0±4.6 | 95.0±1.6 | 54.0±3.9 | 89.5±2.3 | 91.0±2.7 |
| Yager(p=0.5)+Sig | 94.0±1.8 | 78.0±2.7 | 54.0±4.2 | 47.0±4.0 | 96.0±1.2 | 59.0±3.1 | 89.0±2.1 | 91.0±2.4 |
| Yager(p=2)+S | **100.0±0.0** | **100.0±0.0** | 96.7±1.4 | 91.8±2.3 | **97.5±1.2** | **100.0±0.0** | 85.6±2.9 | **100.0±0.0** |
| Yager(p=2)+R | **100.0±0.0** | 98.0±0.7 | 96.5±0.9 | 90.0±1.9 | 94.6±1.2 | **100.0±0.0** | 89.4±1.9 | **100.0±0.0** |
| Yager(p=2)+Sig | **100.0±0.0** | 99.0±0.6 | 93.0±1.6 | 92.0±1.8 | 95.1±1.5 | **100.0±0.0** | 90.4±2.0 | **100.0±0.0** |
| Hamacher+S | **100.0±0.0** | 96.7±1.3 | 99.2±0.8 | 94.4±1.7 | 93.7±1.8 | **100.0±0.0** | 90.4±2.2 | 98.8±1.0 |
| Hamacher+R | 99.0±1.0 | 95.0±1.4 | 96.5±1.0 | 92.7±1.8 | 93.9±1.5 | **100.0±0.0** | 89.4±1.9 | 99.1±0.5 |
| Hamacher+Sig | 99.0±1.0 | 94.0±1.9 | 95.5±1.2 | 93.8±1.6 | 92.9±1.3 | **100.0±0.0** | 87.5±2.1 | 96.2±1.2 |
| ***Adaptive Dual-Loss*** | | | | | | | | |
| Gödel+S | **100.0±0.0** | 84.2±2.8 | **99.3±0.2** | 76.8±3.5 | 95.4±1.6 | **100.0±0.0** | 79.7±3.3 | 95.1±1.7 |
| Gödel+R | 96.0±1.5 | 92.0±1.8 | 96.0±1.2 | 90.5±1.9 | 94.0±1.6 | **100.0±0.0** | 86.0±2.4 | **100.0±0.0** |
| Gödel+Sig | 99.0±0.9 | 94.5±1.6 | 96.0±1.1 | 90.0±1.7 | 89.5±1.8 | **100.0±0.0** | 87.0±2.0 | **100.0±0.0** |
| Product+S | 94.6±1.7 | **99.0±0.9** | **99.6±0.1** | 84.1±2.9 | 84.7±3.0 | 98.2±0.8 | 95.8±1.5 | 97.4±1.3 |
| Product+R | 98.0±1.3 | 90.0±1.6 | 91.0±1.7 | 84.5±1.9 | 83.0±2.3 | 98.5±0.7 | 92.0±1.8 | 99.0±0.6 |
| Product+Sig | 99.0±1.0 | 89.0±1.8 | 90.0±1.6 | 83.0±2.1 | 83.5±2.2 | 97.5±0.9 | 83.0±2.6 | 98.5±0.5 |
| Łukasiewicz+S | 93.2±1.9 | 77.8±3.2 | 74.9±3.6 | **96.7±1.4** | 90.6±2.5 | 85.9±2.9 | 82.7±3.2 | 92.3±2.1 |
| Łukasiewicz+R | 99.0±1.0 | 80.0±2.9 | 92.0±2.0 | 89.0±2.1 | 90.0±2.0 | 87.0±1.9 | 88.5±2.2 | 93.0±1.7 |
| Łukasiewicz+Sig | 99.0±1.0 | 83.0±2.6 | 92.0±1.8 | 89.0±2.4 | 88.0±1.9 | 82.0±2.4 | 85.0±2.4 | 94.0±1.6 |
| Yager(p=0.5)+S | 82.0±2.4 | 70.0±3.1 | 84.0±2.2 | 90.0±2.0 | 89.0±2.1 | 98.0±0.6 | 76.0±3.5 | 99.0±0.3 |
| Yager(p=0.5)+R | 86.0±1.9 | 72.0±2.7 | 83.0±2.4 | 89.0±1.8 | 90.0±1.7 | 99.0±0.4 | 78.5±2.2 | 99.0±0.4 |
| Yager(p=0.5)+Sig | 81.0±2.1 | 70.5±2.8 | 81.0±2.7 | 90.0±2.2 | 88.0±1.8 | 98.0±0.7 | 77.0±2.9 | 99.0±0.3 |
| Yager(p=2)+S | **100.0±0.0** | 93.0±1.4 | 98.0±0.6 | 96.0±1.2 | 94.0±1.4 | **100.0±0.0** | 88.0±1.9 | **100.0±0.0** |
| Yager(p=2)+R | **100.0±0.0** | 90.0±1.5 | 97.0±0.8 | 93.0±1.6 | 93.0±1.5 | **100.0±0.0** | 86.5±1.7 | **100.0±0.0** |
| Yager(p=2)+Sig | **100.0±0.0** | 93.5±1.2 | 97.0±0.7 | 94.0±1.3 | 93.5±1.4 | **100.0±0.0** | 89.5±1.6 | **100.0±0.0** |
| Hamacher+S | 97.4±1.3 | 96.8±1.5 | 95.0±1.2 | 93.1±1.9 | **97.4±1.2** | 98.1±0.9 | 91.3±2.4 | 97.8±0.8 |
| Hamacher+R | 91.0±2.0 | 84.0±2.6 | 94.0±1.5 | 90.0±2.0 | 94.0±1.8 | 90.0±2.3 | 82.0±2.5 | 97.0±1.2 |
| Hamacher+Sig | 90.0±1.9 | 86.0±2.4 | 94.0±1.3 | 90.0±1.7 | 92.0±1.6 | 92.0±1.9 | 84.0±1.8 | 98.0±0.9 |
| ***Baselines*** | | | | | | | | |
| LTN | **100.0±0.0** | 90.3±2.4 | 89.7±2.6 | 72.6±3.9 | 94.8±1.7 | 46.7±4.9 | **98.1±1.2** | 92.1±2.2 |
| F$\mathcal{ALCON}$ | **100.0±0.0** | 76.4±3.3 | 83.9±3.1 | 62.7±4.1 | 91.8±2.3 | 47.1±5.0 | 97.6±1.3 | 91.6±2.3 |
| BoxEL | 91.4±2.5 | 90.2±2.4 | 89.6±2.7 | 61.3±4.2 | 94.9±1.6 | 47.2±4.3 | 84.8±3.1 | 92.1±2.2 |
| Box2EL | **100.0±0.0** | 77.1±3.2 | 93.2±2.1 | 59.1±4.3 | 93.7±1.8 | 52.4±4.1 | 84.9±3.0 | 91.6±2.3 |
| ELEmbedding | **100.0±0.0** | 78.9±3.1 | 88.3±2.8 | 67.1±4.0 | **99.7±0.1** | 50.2±4.6 | 93.6±2.0 | 95.8±1.4 |
| CatE | 99.0±1.0 | 86.0±2.6 | 89.0±1.8 | 73.0±3.2 | 92.5±1.6 | 80.0±2.9 | 95.0±1.3 | 94.0±1.6 |
| DeepProbLog | 95.5±0.9 | 81.0±3.0 | 91.0±1.9 | 70.0±2.7 | 91.0±1.8 | 80.0±2.6 | 94.0±1.4 | 90.0±1.8 |
| DeepStochLog | **100.0±0.0** | 86.0±2.8 | 91.5±1.7 | 71.0±2.9 | 90.0±2.0 | 85.0±2.4 | 94.0±1.5 | 93.0±1.6 |
| NeurASP | **100.0±0.0** | 87.0±2.7 | 93.5±1.4 | 75.0±2.6 | 87.5±2.3 | 72.0±2.8 | 91.0±1.8 | 90.0±1.5 |

Table 5: Performance comparison on MAR Task with 80% mask rate across eight ontology datasets. Results show success rates (%) with standard deviations. Models are categorized into five groups: **Hierarchical Loss Models** (with S-/R-/Sigmoidal implications), **Rule-based Loss Models**, **Adaptive Dual-Loss Models** (dynamic $\omega_R$ based on quantifier prevalence), and **Baseline Models**. Best performance per dataset is shown in **bold**, second-best underlined.

| Mask Rate / Model | Family | Family2 | Glycan | Glyco | Nifdys | Nihss | Ontodm | Sso |
|---|---|---|---|---|---|---|---|---|
| **Mask Rate 0.8** | | | | | | | | |
| *Hierarchy Models* | | | | | | | | |
| Gödel+S | 77.8±1.7 | 62.4±3.1 | 93.7±1.9 | 87.6±2.7 | 76.9±3.2 | **100.0±0.0** | 58.7±4.1 | 97.8±0.8 |
| Gödel+R | 90.6±2.1 | 93.8±2.7 | 83.4±3.2 | 88.7±1.9 | 92.3±2.1 | **100.0±0.0** | 61.2±4.4 | **100.0±0.0** |
| Gödel+Sig | 82.5±2.6 | 77.5±3.4 | 89.5±2.1 | 84.0±3.0 | 78.0±3.3 | **100.0±0.0** | 62.0±3.8 | 98.5±0.9 |
| Product+S | 94.9±2.4 | 71.8±1.9 | 91.5±1.8 | 88.3±2.6 | 87.6±2.8 | **100.0±0.0** | 75.4±3.1 | **100.0±0.0** |
| Product+R | **100.0±0.0** | 73.7±3.1 | 97.2±1.4 | **95.9±0.3** | 92.1±1.4 | **100.0±0.0** | 83.6±2.5 | **100.0±0.0** |
| Product+Sig | 96.0±1.8 | 70.0±2.9 | 92.0±1.9 | 89.0±2.4 | 85.0±3.1 | **100.0±0.0** | 76.0±3.5 | 99.0±0.6 |
| Łukasiewicz+S | 78.2±2.4 | 69.1±2.7 | 92.0±1.1 | 87.8±2.3 | 86.9±2.9 | **100.0±0.0** | 64.3±1.8 | **100.0±0.0** |
| Łukasiewicz+R | **100.0±0.0** | 71.8±2.4 | 97.3±0.6 | **96.4±1.7** | 92.4±2.0 | **100.0±0.0** | 92.9±1.5 | **100.0±0.0** |
| Łukasiewicz+Sig | 99.0±1.0 | 63.0±3.6 | 92.0±2.0 | 89.0±2.8 | 88.0±3.0 | 97.0±0.9 | 78.0±3.1 | 98.0±0.9 |
| Yager(p=0.5)+S | 88.0±2.1 | 59.0±3.8 | 78.0±3.2 | 84.0±3.0 | 86.0±2.8 | **100.0±0.0** | 71.0±4.2 | 97.0±1.0 |
| Yager(p=0.5)+R | 87.0±2.2 | 61.5±3.6 | 80.0±3.5 | 86.0±2.5 | 92.0±1.8 | **100.0±0.0** | 90.0±2.2 | 97.0±0.9 |
| Yager(p=0.5)+Sig | 88.5±2.0 | 63.5±3.3 | 82.0±3.0 | 84.0±3.1 | 86.5±2.4 | 99.5±0.3 | 73.0±3.9 | 99.0±0.5 |
| Yager(p=2)+S | 100.0±0.0 | 73.0±3.0 | 89.0±1.9 | 89.0±2.6 | 83.0±3.0 | **100.0±0.0** | 76.0±3.4 | 99.0±0.7 |
| Yager(p=2)+R | 100.0±0.0 | 69.0±3.3 | 95.0±1.0 | 92.0±1.8 | 90.0±2.0 | **100.0±0.0** | 87.5±2.0 | **100.0±0.0** |
| Yager(p=2)+Sig | 100.0±0.0 | 71.5±3.0 | 92.0±1.3 | 92.0±1.9 | 92.0±1.7 | **100.0±0.0** | 80.0±2.6 | **100.0±0.0** |
| Hamacher+S | 83.9±4.6 | 40.2±5.2 | 94.4±1.5 | 92.3±1.8 | 80.7±4.1 | **100.0±0.0** | 73.1±3.6 | **100.0±0.0** |
| Hamacher+R | 91.7±2.0 | 95.1±1.1 | **100.0±0.0** | 95.4±1.7 | 94.4±1.9 | **100.0±0.0** | 76.2±2.5 | **100.0±0.0** |
| Hamacher+Sig | 85.7±3.5 | 83.0±3.2 | 92.0±1.8 | 91.0±2.2 | 88.0±3.0 | 96.5±1.5 | 75.0±3.0 | 97.0±1.0 |
| *Rule-Based Models* | | | | | | | | |
| Gödel+S | **100.0±0.0** | **100.0±0.0** | 95.2±1.8 | 87.8±2.7 | 91.3±2.4 | **100.0±0.0** | 87.2±2.9 | **100.0±0.0** |
| Gödel+R | 98.0±1.3 | 94.0±2.2 | 92.0±2.4 | 87.5±2.3 | 90.5±2.0 | 99.6±0.4 | 84.0±2.8 | 99.0±0.6 |
| Gödel+Sig | 99.0±0.9 | 92.0±2.0 | 93.5±1.9 | 88.0±2.1 | 91.5±1.9 | 99.9±0.2 | 89.0±2.2 | **100.0±0.0** |
| Product+S | **100.0±0.0** | 77.4±3.4 | 96.1±1.5 | 90.3±2.5 | 90.4±2.6 | **100.0±0.0** | 83.8±3.3 | **100.0±0.0** |
| Product+R | 99.0±1.0 | 84.0±2.6 | 94.0±1.6 | 89.0±2.2 | 91.5±1.9 | 99.0±0.5 | 85.5±2.1 | **100.0±0.0** |
| Product+Sig | 98.5±1.1 | 79.0±3.0 | 95.5±1.4 | 89.5±2.1 | 89.5±2.2 | 99.6±0.4 | 82.5±2.8 | 99.5±0.4 |
| Łukasiewicz+S | **100.0±0.0** | 68.9±3.7 | 60.4±4.4 | 50.7±4.8 | 85.4±3.2 | 65.1±4.3 | 83.2±3.5 | 90.3±2.6 |
| Łukasiewicz+R | 99.0±1.0 | 71.0±3.3 | 64.0±3.9 | 52.0±4.5 | 88.0±2.4 | 77.0±3.4 | 78.0±3.2 | 89.5±3.0 |
| Łukasiewicz+Sig | 99.0±1.0 | 70.0±3.1 | 58.0±4.2 | 57.5±4.8 | 88.5±2.2 | 67.0±3.8 | 86.0±2.9 | 91.0±2.1 |
| Yager(p=0.5)+S | 90.2±2.6 | 77.3±3.3 | 45.6±4.9 | 44.1±5.0 | 95.4±1.6 | 46.0±5.1 | 88.7±2.8 | 90.1±2.7 |
| Yager(p=0.5)+R | 94.0±1.8 | 78.5±2.9 | 53.0±4.7 | 47.0±4.8 | 93.8±1.8 | 52.5±4.0 | 87.5±2.6 | 89.8±2.8 |
| Yager(p=0.5)+Sig | 92.5±1.9 | 76.0±3.0 | 51.5±4.3 | 46.0±4.3 | 94.5±1.5 | 58.0±3.3 | 87.0±2.5 | 89.5±2.5 |
| Yager(p=2)+S | **100.0±0.0** | **100.0±0.0** | 95.3±1.7 | 90.4±2.6 | **97.5±1.2** | **100.0±0.0** | 84.2±3.2 | **100.0±0.0** |
| Yager(p=2)+R | **100.0±0.0** | 98.2±0.9 | 95.8±1.0 | 89.8±2.1 | 94.0±1.4 | **100.0±0.0** | 88.0±2.0 | **100.0±0.0** |
| Yager(p=2)+Sig | **100.0±0.0** | 98.8±0.7 | 92.0±1.6 | 91.5±1.9 | 95.0±1.5 | **100.0±0.0** | 89.0±2.1 | **100.0±0.0** |
| Hamacher+S | **100.0±0.0** | 93.4±1.9 | 96.5±1.4 | 91.2±2.4 | 90.8±2.5 | **100.0±0.0** | 87.4±2.9 | 95.9±1.6 |
| Hamacher+R | 99.0±1.0 | 92.0±1.6 | 95.0±1.3 | 90.0±2.0 | 92.5±1.8 | **100.0±0.0** | 86.0±2.3 | 97.0±1.0 |
| Hamacher+Sig | 99.0±1.0 | 90.0±2.0 | 94.0±1.5 | 90.5±1.8 | 91.0±1.7 | **100.0±0.0** | 85.0±2.0 | 95.0±1.2 |
| *Adaptive Dual-Loss* | | | | | | | | |
| Gödel+S | **100.0±0.0** | 82.7±3.1 | **99.3±0.2** | 75.3±3.8 | 94.1±1.9 | **100.0±0.0** | 78.3±3.6 | 93.8±2.0 |
| Gödel+R | 94.0±1.9 | 90.5±2.0 | 93.5±1.8 | 88.8±2.2 | 92.0±1.9 | **100.0±0.0** | 84.5±2.6 | 98.5±0.8 |
| Gödel+Sig | 98.5±1.0 | 92.0±1.8 | 94.0±1.3 | 88.5±2.0 | 88.0±1.9 | **100.0±0.0** | 85.0±2.2 | 99.0±0.6 |
| Product+S | 93.1±2.0 | **99.0±0.9** | **99.6±0.1** | 82.7±3.2 | 83.3±3.3 | 96.9±1.1 | **95.8±1.5** | 96.1±1.6 |
| Product+R | 96.0±1.4 | 88.5±1.9 | 90.0±1.9 | 82.5±2.1 | 81.5±2.6 | 96.0±1.2 | 90.0±2.0 | 97.5±0.9 |
| Product+Sig | 97.5±1.2 | 86.0±2.1 | 88.5±1.8 | 81.5±2.3 | 81.0±2.4 | 95.0±1.4 | 82.0±2.8 | 96.5±0.8 |
| Łukasiewicz+S | 91.8±2.2 | 76.4±3.5 | 73.6±3.9 | **96.7±1.4** | 89.2±2.8 | 84.5±3.2 | 81.3±3.5 | 91.0±2.4 |
| Łukasiewicz+R | 97.5±1.3 | 78.0±2.8 | 90.0±2.3 | 86.5±2.6 | 88.5±2.1 | 85.0±2.2 | 87.0±2.6 | 91.5±1.9 |
| Łukasiewicz+Sig | 97.0±1.4 | 80.0±2.9 | 89.0±2.0 | 86.0±2.8 | 86.0±1.9 | 80.0±2.5 | 82.5±2.6 | 92.0±1.8 |
| Yager(p=0.5)+S | 80.0±2.6 | 68.0±3.4 | 82.0±2.3 | 88.5±2.4 | 87.5±2.5 | 97.5±0.7 | 74.5±3.6 | 98.5±0.5 |
| Yager(p=0.5)+R | 84.0±2.1 | 70.0±2.9 | 81.0±2.6 | 87.5±2.0 | 88.0±1.8 | 98.5±0.5 | 76.0±2.3 | 98.5±0.5 |
| Yager(p=0.5)+Sig | 79.5±2.3 | 69.0±3.0 | 79.0±2.9 | 87.0±2.6 | 86.0±1.8 | 97.5±0.8 | 75.0±2.8 | 98.5±0.4 |
| Yager(p=2)+S | **100.0±0.0** | 91.5±1.6 | 97.5±0.6 | 95.0±1.3 | 93.0±1.6 | **100.0±0.0** | 86.5±1.9 | **100.0±0.0** |
| Yager(p=2)+R | **100.0±0.0** | 88.5±1.8 | 96.0±0.8 | 92.5±1.7 | 92.0±1.6 | **100.0±0.0** | 84.0±1.8 | **100.0±0.0** |
| Yager(p=2)+Sig | **100.0±0.0** | 90.5±1.5 | 95.5±0.8 | 93.0±1.4 | 92.5±1.5 | **100.0±0.0** | 86.0±1.7 | **100.0±0.0** |
| Hamacher+S | 96.0±1.6 | 95.4±1.8 | 95.9±1.7 | 91.7±2.2 | **97.4±1.2** | 96.8±1.2 | 90.0±2.7 | 96.5±1.1 |
| Hamacher+R | 92.0±1.9 | 84.5±2.7 | 93.5±1.6 | 89.0±2.0 | 94.0±1.8 | 90.5±2.1 | 83.0±2.6 | 95.0±1.4 |
| Hamacher+Sig | 91.0±1.8 | 86.0±2.3 | 93.0±1.5 | 89.5±1.7 | 92.0±1.6 | 91.5±1.9 | 85.0±1.9 | 96.0±1.0 |
| *Baselines* | | | | | | | | |
| LTN | **100.0±0.0** | 88.9±2.7 | 88.3±2.9 | 71.2±4.2 | 93.4±2.0 | 45.3±5.2 | **98.1±1.2** | 90.8±2.5 |
| F𝒜ℒ𝒞ON | **100.0±0.0** | 75.1±3.6 | 82.4±3.4 | 61.4±4.4 | 90.5±2.6 | 45.8±5.3 | 97.6±1.3 | 90.2±2.6 |
| BoxEL | 90.0±2.8 | 88.8±2.7 | 88.2±3.0 | 60.0±4.5 | 93.6±1.9 | 46.0±4.6 | 83.4±3.4 | 90.8±2.5 |
| Box2EL | **100.0±0.0** | 75.8±3.5 | 91.8±2.4 | 57.8±4.6 | 92.3±2.1 | 51.1±4.4 | 83.5±3.3 | 90.2±2.6 |
| ELEmbedding | **100.0±0.0** | 77.4±3.4 | 86.9±3.1 | 65.8±4.3 | **99.7±0.1** | 49.0±4.9 | 92.2±2.3 | 94.4±1.7 |
| CatE | 97.0±1.5 | 84.0±2.9 | 87.0±2.1 | 71.0±3.6 | 91.5±1.8 | 78.0±3.1 | 93.0±1.7 | 92.0±1.9 |
| DeepProbLog | 92.0±1.8 | 79.5±3.2 | 89.0±2.0 | 68.0±2.8 | 90.0±1.9 | 78.5±2.9 | 92.0±1.9 | 88.5±1.8 |
| DeepStochLog | 99.0±1.0 | 84.0±3.0 | 89.5±1.9 | 69.0±3.1 | 88.5±2.2 | 83.0±2.6 | 92.0±1.8 | 91.0±1.7 |
| NeurASP | 99.0±1.0 | 85.0±2.9 | 92.0±1.6 | 73.5±2.8 | 86.0±2.4 | 70.0±3.0 | 89.0±1.9 | 88.0±1.6 |

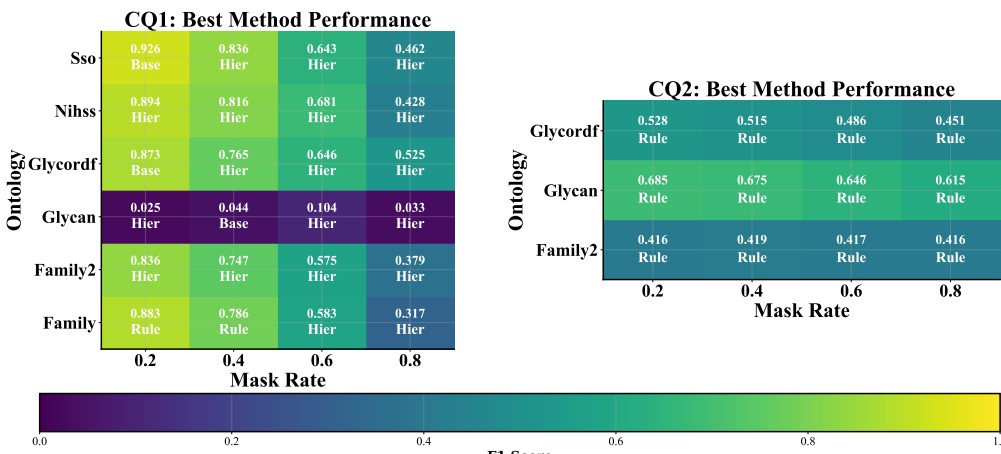

Figure 4: Performance heatmap on CQA tasks. Values represent F1-scores averaged over 5 independent runs. Cell colors indicate the top-performing model category (Baseline, Rule-based, or Hierarchical) for each ontology-mask rate combination.

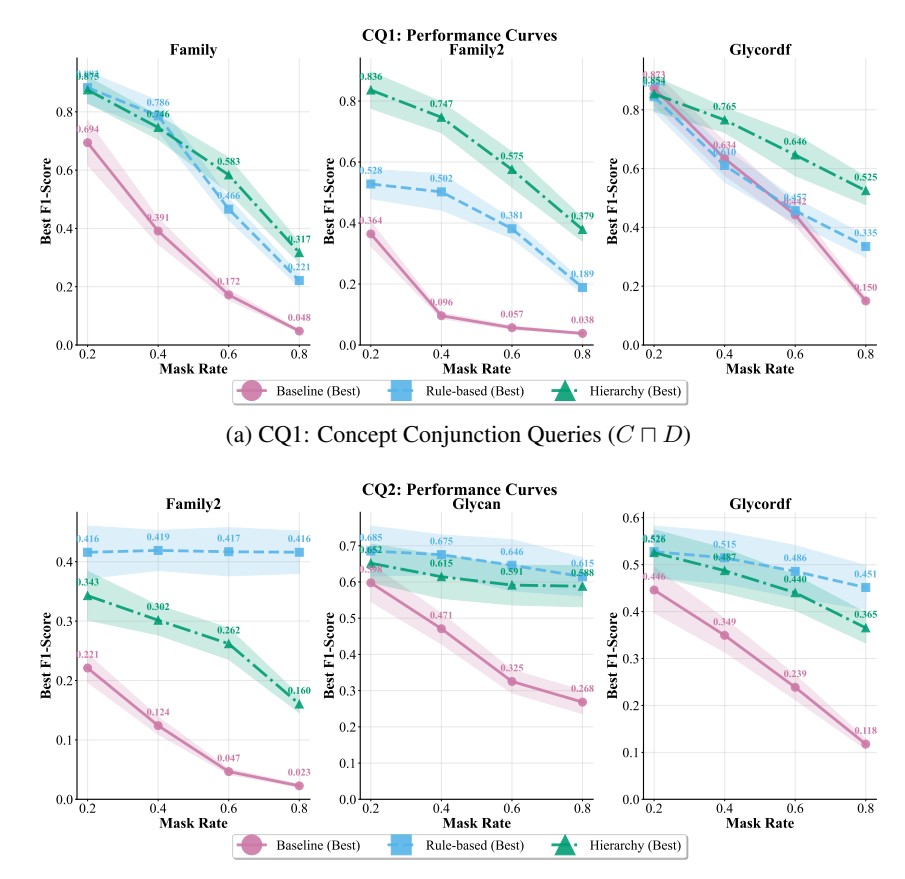

(a) CQ1: Concept Conjunction Queries ($C \sqcap D$)

(b) CQ2: Role-based Queries ($C \sqcap \exists r.D$)

Figure 5: Best F1-score achieved within each model category across mask rates for test ontologies. Each data point represents the mean of 5 independent runs; error bars denote one standard deviation.

### D.3 TASK 3: CONJUNCTIVE QUERY ANSWERING (CQA)

This task assesses the model's ability to preserve fine-grained semantic relationships after the revision process (**RQ2**). While the MAR task measures global axiom satisfaction, CQA evaluates whether the learned interpretation can correctly answer complex logical queries—a capability that demands meaningful embeddings for both concepts and roles, not merely constraint satisfaction.

#### D.3.1 EXPERIMENTAL SETUP

After learning the revised grounding $\mathcal{I}''$ from the MAR task, we evaluate its quality using conjunctive query answering. For each ontology, we automatically generate **100 queries** spanning two structurally different forms:

- **CQ1: (Concept Conjunctions)** of the form $C \sqcap D$, testing basic concept intersection.
- **CQ2: (Role-based Queries)** of the form $C \sqcap \exists r.D$, requiring multi-step reasoning over roles.

Answers are extracted from $\mathcal{I}''$ using a membership threshold of $0.8$ and evaluated against ground-truth answers derived from the ideal grounding $\mathcal{I}$. We report **Precision**, **Recall**, and **F1-score** as evaluation metrics.

#### D.3.2 RESULTS AND ANALYSIS

As shown in Figures 4 and 5, NeSy$\mathcal{ALC}$ significantly outperforms all baselines on F1-score across most test cases, demonstrating that our method preserves the fine-grained semantic structure required for multi-step reasoning.

**Loss Strategy Specialization.** A notable finding is the clear specialization of our proposed loss strategies. The **Hierarchical Loss** consistently achieved the best results on CQ1 (concept conjunctions), aligning with its design to enforce global ontological consistency. Conversely, the **Rule-Based Loss** excelled on CQ2 (role-based queries), as its semantically-gated mechanism effectively handles the subtleties of quantifiers and mitigates representational collapse. This complementary behavior validates the design rationale behind both strategies.

**Impact of Query Complexity and Ontological Expressivity.** NeSy$\mathcal{ALC}$ demonstrates superior performance across most scenarios, particularly at lower masking rates where logical constraints can effectively guide the revision process. A consistent pattern emerges: our framework excels on concept conjunction queries while facing greater challenges with role-based queries. This performance gap stems from the increased semantic complexity of role interpretation, which requires more precise groundings to maintain the delicate balance between existential and universal quantification semantics.

The ontological characteristics in Table 2 reveal important factors influencing performance. Ontologies with richer expressivity (containing universal quantifiers, disjunctions, and complex negation patterns) benefit more from NeSy$\mathcal{ALC}$'s full $\mathcal{ALC}$ support, while simpler taxonomic structures show smaller performance differentials. Notably, role-based queries cannot be generated for the `Sso` ontology due to its restricted expressivity profile, highlighting the dependency between ontological richness and evaluable query patterns.

Additionally, several baselines exhibit poor performance not due to algorithmic limitations but because they fail to effectively revise masked information, resulting in membership predictions concentrated in the uncertain range $[0.2, 0.8]$, which our crisp transformation interprets as unknown.

**Robustness to Ontology Imperfections.** To provide deeper insights into NeSy$\mathcal{ALC}$'s behavior with imperfect knowledge, we conducted experiments on two types of challenging data: (i) inherently inconsistent ontologies (`Ontodm` and `Nifdys`), and (ii) manually degraded, low-quality versions of consistent ontologies.

Table 6 presents a controlled comparison for `GlycoRDF`, where we deliberately introduced contradictory assertions to simulate real-world knowledge base imperfections. The results reveal a nuanced performance profile: queries targeting inconsistent portions experience significant degradation (precision drops 17–30% across masking rates), as classical DL reasoners used for ground truth fail on logical contradictions. However, NeSy$\mathcal{ALC}$ maintains competitive performance on queries targeting

Table 6: Comparison between the original consistent `GlycoRDF` and a version with deliberately introduced contradictions. The baseline row shows absolute Precision (P) and Recall (R) values; the inconsistent row shows performance degradation as percentage point differences.

| Ontology Version | Mask 0.2 | | Mask 0.4 | | Mask 0.6 | | Mask 0.8 | |
|---|---|---|---|---|---|---|---|---|
| | P (%) | R (%) | P (%) | R (%) | P (%) | R (%) | P (%) | R (%) |
| Consistent (Baseline) | 87.0 | 61.3 | 82.6 | 41.6 | 69.6 | 31.7 | 65.2 | 19.9 |
| Inconsistent ($\Delta$) | −26.1 | −36.8 | −30.4 | −22.0 | −30.5 | −16.8 | −17.4 | −7.6 |

consistent portions, demonstrating robust compartmentalization. This graceful degradation—rather than catastrophic failure—is a valuable property for real-world applications where perfect logical consistency is rare.

Table 7: Transformation techniques to simulate real-world ontology imperfections. Each transformation targets a specific type of knowledge degradation commonly encountered in practical applications.

| Transformation | Operation | Example |
|---|---|---|
| **TBox Modifications** | | |
| Strengthen Constraints | Alter | Malignant $\sqsubseteq$ Tumor $\rightarrow$ Malignant $\sqsubseteq$ Tumor $\sqcap$ $\exists$invades.Tissue |
| Weaken Constraints | Alter | Tumor $\equiv$ Mass $\sqcap$ Abnormal $\rightarrow$ Tumor $\sqsubseteq$ Mass $\sqcap$ Abnormal |
| Remove Disjointness | Remove | Adenocarcinoma $\sqcap$ SquamousCell $\equiv$ $\bot$ |
| Remove Concept Definition | Remove | Glioblastoma $\equiv$ Glioma $\sqcap$ HighGrade $\sqcap$ $\forall$shows.Necrosis |
| Blur Subclass Relation | Alter | Astrocytoma $\sqsubseteq$ Glioma $\rightarrow$ Astrocytoma $\sqsubseteq$ GlialTumor $\sqsubseteq$ Glioma |
| **ABox Modifications** | | |
| Remove ABox Assertions | Remove | (patient27, tumor35) : hasDiagnosis |
| Add Underdefined Instances | Add | patient42 : Patient *(without diagnosis, symptoms, or test results)* |

Table 7 details seven transformation techniques used to create low-quality ontology variants, simulating common real-world imperfections such as weakened constraints, missing definitions, and incomplete instance data.

Table 8: Performance degradation on low-quality ontology variants. Values show absolute percentage point differences in Precision (P) and Recall (R) compared to the baseline. Green indicates minor degradation (<10%); red indicates severe degradation (≥20%).

| Transformation | Mask 0.2 | | Mask 0.4 | | Mask 0.6 | | Mask 0.8 | |
|---|---|---|---|---|---|---|---|---|
| | $\Delta$P | $\Delta$R | $\Delta$P | $\Delta$R | $\Delta$P | $\Delta$R | $\Delta$P | $\Delta$R |
| **Baseline (Absolute)** | 77.0 | 45.7 | 57.7 | 15.3 | 38.5 | 7.2 | 15.4 | 2.3 |
| *TBox Degradations* | | | | | | | | |
| Weakened Constraints | −7.8 | −8.0 | −3.8 | −0.6 | −30.8 | −6.1 | −7.7 | −0.4 |
| Removed Constraints | −10.1 | −11.3 | −14.2 | −2.8 | −10.3 | −0.8 | −10.1 | −1.2 |
| Removed TBox | −2.4 | −6.5 | −12.4 | −3.3 | −7.7 | −1.6 | −5.1 | −0.7 |
| Redundant TBox | −13.5 | −19.6 | −11.6 | −3.3 | −17.9 | −3.3 | −5.1 | −0.5 |
| Blurred Subclass | −1.5 | −6.2 | −32.1 | −8.9 | −7.7 | −0.1 | ±0.0 | −0.5 |
| *ABox Degradations* | | | | | | | | |
| Removed ABox | −13.7 | −17.0 | −20.1 | −0.3 | −23.1 | −3.8 | −10.1 | −1.4 |
| Redundant ABox | −4.3 | −11.5 | −18.0 | −3.7 | −23.1 | −3.9 | −7.7 | −0.5 |

The results in Table 8 indicate that while performance degrades when removing or altering useful constraints—as expected—the degradation is often minor, suggesting strong capability to preserve useful information even with imperfect knowledge. Notably, when underdefined instances are added, the model correctly assigns them membership values near 0.5 for all classes, representing maximum uncertainty and demonstrating NeSy$\mathcal{ALC}$'s principled handling of unknown information.

# E  HYPERPARAMETER SETTINGS AND EXPERIMENTAL DETAILS

To ensure reproducibility of our experimental results, this section provides a comprehensive overview of the hyperparameter settings for NeSy$\mathcal{ALC}$ and all baseline models. A systematic grid search was conducted for all models across all datasets to identify the optimal configuration. The primary metric for optimization during this search was the **Success Rate** on the MAR task, using a dedicated validation set split from the original ontologies. All reported results are based on these optimal settings, averaged over 5 independent runs with different random seeds to ensure statistical significance.

## E.1  GENERAL TRAINING CONFIGURATION

For all experiments, including our models and all baselines, we used the **Adam optimizer** (Kingma & Ba, 2015). We employed an early stopping strategy with a patience of **10 epochs**, monitoring the validation set's Success Rate to prevent overfitting and select the best-performing model state. All models were implemented in PyTorch (Paszke et al., 2019) and trained on NVIDIA RTX 3090 GPUs. The training time per epoch varied depending on the ontology's size and complexity, ranging from a few seconds for smaller datasets like `Nihss` to several minutes for larger ones like `Nifdys`.

## E.2  HYPERPARAMETER SEARCH SPACE

The grid search for all models was designed to be comprehensive, including the following hyperparameter ranges:

- **Learning Rate (LR):** We explored a standard range of values suitable for deep learning models, specifically $\{1 \times 10^{-5}, 2 \times 10^{-5}, 5 \times 10^{-5}, 1 \times 10^{-4}, 2 \times 10^{-4}, 5 \times 10^{-4}\}$.
- **Embedding Dimension (Dim):** We tested dimensions of $\{100, 200, 400\}$ to assess the impact of representation capacity.
- **Batch Size (BS):** We evaluated batch sizes of $\{64, 128, 256, 512, 1024\}$ to balance computational efficiency and training stability.
- **Fuzzy Operator Parameters (for NeSy$\mathcal{ALC}$):** For the parameterized Yager t-norm family, we searched for the parameter $p \in \{0.5, 1.5, 2.0, 5.0\}$. For clarity in our main results, we display the performance for representative values ($p \in \{0.5, 2.0\}$). For our Adaptive Dual-Loss strategy, the scaling factor $\lambda$ was tuned within the range $(0.5, 10.0)$, and the optimal value was consistently found to be $\lambda = 4.0$ across most datasets, indicating stable behavior for this mechanism.

## E.3  OPTIMAL HYPERPARAMETER CONFIGURATIONS

Tables 9, 10, and 11 detail the optimal hyperparameter settings identified through our grid search. The results cover all eight evaluated ontologies, which we have grouped by their relative scale and complexity to better illustrate emerging patterns. For NeSy$\mathcal{ALC}$, we report configurations for both the R-implication variants (denoted with +R suffix) and the Adaptive Dual-Loss variants for each t-norm. An important finding from our search was that for a given t-norm, the optimal hyperparameters were consistent across its R-implication and Adaptive variants, suggesting that the choice of loss strategy does not significantly alter the model's optimal capacity requirements.

Our analysis of the optimal configurations reveals a clear and logical trend related to dataset characteristics. For the smaller-scale ontologies (`Family`, `Sso`, `Nihss`), models performed best with a relatively small batch size of 64 and a standard embedding dimension of 200. In contrast, the medium-scale ontologies (`Family2`, `Glycan`, `GlycoRDF`), which feature more complex axiom structures, consistently benefited from a larger representational capacity (400 dimensions) and a correspondingly larger batch size of 256. Finally, for the largest and most axiomatically dense ontologies (`Ontodm`, `Nifdys`), optimal performance required more stable training conditions: a slightly lower learning rate of $1 \times 10^{-4}$ and the largest effective batch size of 512.

Table 9: Optimal hyperparameters for all models on the **Family**, **Sso**, and **Nihss** datasets.

| Category | Model | LR | Dim (Emb) | Dim (Hidden) | Dim (Other) | BS |
|---|---|---|---|---|---|---|
| NeSy$\mathcal{ALC}$ | Gödel | $2 \times 10^{-4}$ | 200 | – | – | 64 |
| | Product | $2 \times 10^{-4}$ | 200 | – | – | 64 |
| | Łukasiewicz | $2 \times 10^{-4}$ | 200 | – | – | 64 |
| | Yager(p=0.5) | $2 \times 10^{-4}$ | 200 | – | – | 64 |
| | Yager(p=2) | $2 \times 10^{-4}$ | 200 | – | – | 64 |
| | Hamacher | $2 \times 10^{-4}$ | 200 | – | – | 64 |
| Baselines | LTN | $2 \times 10^{-4}$ | 200 | 256 | – | 64 |
| | EL Embedding | $2 \times 10^{-4}$ | 200 | – | – | 64 |
| | F$\mathcal{ALC}$ON | $2 \times 10^{-4}$ | 200 | – | – | 64 |
| | BoxEL | $2 \times 10^{-4}$ | 200 | – | 50 | 64 |
| | Box2EL | $2 \times 10^{-4}$ | 200 | – | 50 | 64 |
| | CatE | $2 \times 10^{-4}$ | 200 | – | – | 64 |
| | DeepProbLog | $2 \times 10^{-4}$ | 128 | 256 | – | 32 |
| | DeepStochLog | $2 \times 10^{-4}$ | 128 | 256 | – | 32 |
| | NeurASP | $2 \times 10^{-4}$ | 128 | 256 | – | 32 |

Table 10: Optimal hyperparameters for all models on the **Family2**, **Glycan**, and **GlycoRDF** datasets.

| Category | Model | LR | Dim (Emb) | Dim (Hidden) | Dim (Other) | BS |
|---|---|---|---|---|---|---|
| NeSy$\mathcal{ALC}$ | Gödel | $2 \times 10^{-4}$ | 400 | – | – | 256 |
| | Product | $2 \times 10^{-4}$ | 400 | – | – | 256 |
| | Łukasiewicz | $2 \times 10^{-4}$ | 400 | – | – | 256 |
| | Yager(p=0.5) | $2 \times 10^{-4}$ | 400 | – | – | 256 |
| | Yager(p=2) | $2 \times 10^{-4}$ | 400 | – | – | 256 |
| | Hamacher | $2 \times 10^{-4}$ | 400 | – | – | 256 |
| Baselines | LTN | $2 \times 10^{-4}$ | 400 | 256 | – | 256 |
| | EL Embedding | $2 \times 10^{-4}$ | 400 | – | – | 256 |
| | F$\mathcal{ALC}$ON | $2 \times 10^{-4}$ | 400 | – | – | 256 |
| | BoxEL | $2 \times 10^{-4}$ | 400 | – | 50 | 256 |
| | Box2EL | $2 \times 10^{-4}$ | 400 | – | 50 | 256 |
| | CatE | $2 \times 10^{-4}$ | 400 | – | – | 256 |
| | DeepProbLog | $2 \times 10^{-4}$ | 256 | 256 | – | 32 |
| | DeepStochLog | $2 \times 10^{-4}$ | 256 | 256 | – | 32 |
| | NeurASP | $2 \times 10^{-4}$ | 256 | 256 | – | 32 |

Table 11: Optimal hyperparameters for all models on the **Ontodm** and **Nifdys** datasets.

| Category | Model | LR | Dim (Emb) | Dim (Hidden) | Dim (Other) | BS |
|---|---|---|---|---|---|---|
| NeSy$\mathcal{ALC}$ | Gödel | $1 \times 10^{-4}$ | 200 | – | – | 512 |
| | Product | $1 \times 10^{-4}$ | 200 | – | – | 512 |
| | Łukasiewicz | $1 \times 10^{-4}$ | 200 | – | – | 512 |
| | Yager(p=0.5) | $1 \times 10^{-4}$ | 200 | – | – | 512 |
| | Yager(p=2) | $1 \times 10^{-4}$ | 200 | – | – | 512 |
| | Hamacher | $1 \times 10^{-4}$ | 200 | – | – | 512 |
| Baselines | LTN | $1 \times 10^{-4}$ | 200 | 128 | – | 512 |
| | EL Embedding | $1 \times 10^{-4}$ | 200 | – | – | 512 |
| | F$\mathcal{ALC}$ON | $1 \times 10^{-4}$ | 200 | – | – | 512 |
| | BoxEL | $1 \times 10^{-4}$ | 200 | – | 50 | 512 |
| | Box2EL | $1 \times 10^{-4}$ | 200 | – | 50 | 512 |
| | CatE | $1 \times 10^{-4}$ | 200 | – | – | 512 |
| | DeepProbLog | $1 \times 10^{-4}$ | 128 | 256 | – | 32 |
| | DeepStochLog | $1 \times 10^{-4}$ | 128 | 256 | – | 32 |
| | NeurASP | $1 \times 10^{-4}$ | 128 | 256 | – | 32 |

